# Rapid meiotic prophase chromosome movements in *Arabidopsis thaliana* are linked to essential reorganization at the nuclear envelope

Laurence Cromer [1,6], Mariana Tiscareno-Andrade[1,6], Sandrine Lefranc[1], Aurélie Chambon[1], Aurélie Hurel[1], Manon Brogniez[1], Julie Guérin[1], Ivan Le Masson [2], Gabriele Adam[3,4,5], Delphine Charif[1], Philippe Andrey [1] & Mathilde Grelon [1] ✉

Meiotic rapid prophase chromosome movements (RPMs) require connections between the chromosomes and the cytoskeleton, involving SUN (Sad1/UNC-84)-domain-containing proteins at the inner nuclear envelope (NE). RPMs remain significantly understudied in plants, with respect to their importance in the regulation of meiosis. Here, we demonstrate that *Arabidopsis thaliana* meiotic centromeres undergo rapid (up to 500 nm/s) and uncoordinated movements during the zygotene and pachytene stages. These centromere movements are not affected by altered chromosome organization and recombination but are abolished in the double mutant *sun1 sun2*. We also document the changes in chromosome dynamics and nucleus organization during the transition from leptotene to zygotene, including telomere attachment to SUN-enriched NE domains, bouquet formation, and nucleolus displacement, all of which were defective in *sun1 sun2*. These results establish *A. thaliana* as a model species for studying the functional implications of meiotic RPMs and demonstrate the mechanistic conservation of telomere-led RPMs in plants.

During meiosis, maternal and paternal chromosomes recombine and segregate in two consecutive divisions, generating genetically distinct cells with half of the chromosome number of the parental organism. This reduction is mandatory to prepare for the doubling of chromosome numbers that occurs after fertilization. In most organisms, equal separation of chromosomes at the first meiotic division relies on forming bivalents (i.e., connected homologous chromosomes), stably held together by crossovers (COs) and sister chromatid cohesion.

Homologous chromosome recognition and association into bivalents occur during meiotic prophase and require that homologous chromosomes find each other within a crowded nuclear space, avoiding all the nonhomologous partners. This achievement is all the more spectacular because homologous chromosome recognition and pairing occur in a context where chromosomes are attached to the nuclear envelope (NE) by their telomeres[1]. Several mechanisms that could compensate for such hindrance and promote proper homologous

[1]Université Paris-Saclay, INRAE, AgroParisTech, Institute Jean-Pierre Bourgin for Plant Sciences (IJPB), 78000 Versailles, France. [2]Université Paris-Saclay, AgroParisTech, INRAE, UMR Agronomie, 91120 Palaiseau, France. [3]Université Paris-Saclay, CNRS, INRAE, Institute of Plant Sciences Paris-Saclay (IPS2), 91190 Gif sur Yvette, France. [4]Université Evry, Institute of Plant Sciences Paris-Saclay (IPS2), 91190 Gif sur Yvette, France. [5]Université Paris Cité, CNRS, INRAE, Institute of Plant Sciences Paris-Saclay (IPS2), 91190 Gif sur Yvette, France. [6]These authors contributed equally: Laurence Cromer, Mariana Tiscareno-Andrade. ✉e-mail: mathilde.grelon@inrae.fr

recognition have been identified; for example, a high level of homologous recombination allows the establishment of multiple trans-interactions between chromosomes[1]. The transient clustering of telomeres in a limited area of the NE (the"telomere bouquet") has also been proposed to facilitate chromosome homologous recognition[2]. The drastic movement of chromosomes during prophase I, also known as Rapid Prophase Movements (RPMs), is a cellular event that might also promote homologous pairing. RPMs can be rotations and/or oscillations of the meiocyte nuclei as well as rapid and erratic chromosome movements[3–6]. While the movement of chromosomes during meiotic prophase I is a conserved dynamic of meiosis, the nature and duration of these movements are quite variable. In *S. pombe*, the whole nucleus is moved back and forth between the two poles of the cells during prophase I, generating coordinated chromosome movements inside the nucleus[7]. In mice, also, meiotic nuclear rotations confer coordinated and unidirectional motion of chromosomes, which adds up with individual and random movements of the chromosomes[8,9]. In *C. elegans* and *S. cerevisiae*, only individual chromosome movements were observed, sometimes associated with drastic deformation of the NE[10–13].

A standard model for meiotic chromosome movement has been constructed from studies in the species mentioned above. Upon entry into meiotic prophase, chromosome ends (telomeres in general or subtelomeric pairing centers in *C. elegans*[14]) become attached to the NE and connected to the cytoplasmic cytoskeleton thanks to the LINC protein complexes (linker of nucleoskeleton and cytoskeleton). The LINC complex is composed of multimers of SUN (Sad1/UNC-84) and KASH (Klarsicht/ANC-1/Syne-1) proteins that traverse the inner and outer nuclear membranes, respectively, and interact in the nuclear envelope lumen[5,15,16]. KASH domain-containing proteins involved in meiotic RPMs show distant relationships among eukaryotes[17–20]. Likewise, the protein complexes involved in telomere or pairing center attachment to the NE are highly species-specific, as are the ones establishing the connection between the LINC complex and the cytoskeleton (see, for example, refs. 21,22).

The exact function(s) of RPMs still need to be better understood. However, RPMs and meiotic recombination are strongly related. For example, in mice, all chromosome movement regulator mutants also have defects in meiotic double-strand break repair. They accumulate an excess of DMC1 foci, and meiotic progression becomes arrested at zygotene; eventually, meiocytes undergo apoptosis[8,23,24]. Also, in *C. elegans*, defects in chromosome movements are associated with the absence of presynaptic homolog alignment, nonhomologous pairing, self-synapsis, and defects in DSB repair (fragmentation)[25,26]. These defects led to the idea that in *C. elegans*, where pairing and synapsis are recombination independent, chromosome movements allow homology assessment and participate in triggering SC polymerization[26]. Data obtained in *S. cerevisiae* and *S. pombe* suggest that chromosome movements could both promote interactions between distantly located DNA molecules (either homologous or nonhomologous, thereby facilitating interaction between chromosomes) and participate in removing unwanted interactions such as ectopic nonhomologous recombination intermediates[11,20,27–30]. This hypothesis agrees that in *C. elegans*, both increasing and decreasing chromosome movement intensity cause meiotic defects[3].

Evidence for such chromosome movements is scarce in plants since it has only been described in maize by tracking DAPI-dense chromosomal regions[31]. Since then, there has been no further investigation of chromosome movement in meiotic prophase despite its likely essential role in recombination. There is no precise description in plants of the RPMs nor of any machinery components involved in chromosome movement. We took the opportunity of recent advances in live imaging of meiosis in *A. thaliana*[32–34] to develop an approach to analyze meiotic prophase centromere dynamics in this model plant. Using 3D fluorescence time-lapse microscopy on male reproductive organs (anthers) and quantitative image analyses, we describe for the first time *A. thaliana* RPMs. We show that prophase centromere movements occur during zygotene and pachytene and that they are not affected by mutations impairing either chromosome axis, SC transverse filament, or meiotic recombination. However, they are completely dependent on SUN1 and SUN2, the two SUN domain-containing proteins shown to be involved in meiosis[35].

## Results

### Live imaging of meiosis to study prophase chromosome movements

Live-cell imaging techniques have been recently applied to *A. thaliana* meiosis but, until now, have mainly been used to study meiotic chromosome segregation[32–34]. Here, we applied the approach described by ref. 33 to analyze meiotic prophase dynamics on whole anthers. We combined two markers, the REC8 protein fused to the RFP (which labels specifically the meiocytes[36]) and the centromeric histone CENH3 protein fused to the GFP at its N-terminus[37], to label and track centromeres (Fig. 1, Supplementary Movies 1 and 2). The REC8-RFP signal faded away quickly but allowed fast and accurate identification of the meiocyte compartment. The GFP-CENH3 signal, however, allowed centromere tracking for several hours with only limited bleaching. Due to the potential phototoxicity and/or tissue death resulting from the removal of the anthers from the flowers, we principally analyzed acquisitions from 2 to 12 min. Centromere tracking of a whole locule (encompassing centromeres from meiocytes and somatic tissues) (Fig. 1 and Supplementary Movies 1 and 2) revealed how meiotic and somatic centromeres show different dynamics (Fig. 1, and below for quantifications).

Meiotic prophase I is divided in five substages: leptotene, zygotene, pachytene, diplotene, and diakinesis. They can be discriminated based on landmarks such as centromere position, centromere number, cell shape, the position of the nucleolus, progression of synapsis and recombination. In the frame of live imaging acquisitions, we have access to only a limited number of these parameters: cell shape, nucleolar position, centromere number, and location. To characterize the movement of chromosomes on anthers for which we were certain of the stages, we successively performed live imaging acquisition and DAPI staining of spread meiotic chromosomes on individual anthers (Fig. 2a and Supplementary Movies 3–6). We observed that square-shaped cells, with a central nucleolus position, always corresponded to anthers at the leptotene stage based on DAPI staining (Fig. 2ai). In these cells, numerous centromeres (close to 10) were present at the periphery of the meiocytes (Supplementary Movie 3). Cells with a peripheral nucleolus and a trapezoidal shape corresponded either to zygotene or pachytene, discriminated only after DAPI staining (Fig. 2aii and iii). In these cells, 3 to 9 GFP-CENH3 signals were identified in the nucleoplasm (Fig. 2a and Supplementary Movies 4, 5). Lastly, round-shaped cells with a peripheral nucleolus corresponded to diplotene (Fig. 2aiv) or diakinesis. In these cells, the number of distinguishable centromeres was comparable to that of zygotene and pachytene. These data show that we can use brightfield images to stage living anthers, but that only subsequent DAPI-staining allows discrimination between zygotene and pachytene stages.

### *A. thaliana* rapid centromere movements start at the zygotene stage and cease at diplotene

Based on live imaging followed by DAPI anther staging, we determined centromeric movement at the different meiotic prophase stages (see above). At the leptotene stage, the displacement of the GFP-CENH3 foci was limited and located close to the nuclear membrane ($n = 15$ cells) (Fig. 2ai, Supplementary Movie 3). In zygotene and pachytene, however, the centromeres usually displayed important displacement in the nucleoplasm ($n = 21$ and $n = 24$ cells for zygotene and pachytene

respectively) (Fig. 2aii, iii, Supplementary Movies 4 and 5). Then, at diplotene, the centromere trajectories became stationary again (n = 15 cells) (Fig. 2aiv, Supplementary Movie 6). To corroborate these observations, we performed a quantitative analysis of individual centromere 3D tracks, following a correction for sample drift (see Materials and Methods). Computing the instant displacement speed revealed large fluctuations along individual tracks (Fig. 2b and Supplementary Fig. 1). Average speed measurements confirmed the absence of specific movement at the leptotene stage, as we did not observe a difference between meiotic (55 ± 20 nm/s, n = 130 tracks) and somatic (46 ± 17 nm/s, n = 34) nuclei (Fig. 2c, Supplementary Fig. 2). This contrasted with the significant difference observed at the subsequent zygotene and pachytene stages, where the average speed of 106 ± 42 nm/s (n = 96) and 100 ± 36 nm/s (n = 100) exhibited a nearly twofold increase compared with the leptotene stage, with no difference between zygotene and pachytene (mixed-effects model, with anther and cell as nested random factors, P = 0.73). At the diplotene stage, the average speed (62 ± 22 nm/s, n = 85) decreased toward the basal level observed in somatic nuclei (40 ± 15 nm/s, n = 54), showing a slowing down of centromeric movements as meiosis progressed. We obtained similar ratios across stages when we considered minimal and maximal speeds instead of averages along tracks (min–max range was: leptotene: 3–250 nm/s; zygotene: 7–425 nm/s; pachytene: 8–492 nm/s; diplotene: 2–285 nm/s). These findings highlight the dynamics and stage specificity of centromeric movements across meiosis.

We next computed the distribution of the turning angle (TA), a descriptor of instant displacement along centromere trajectories. TA is the deflection angle between consecutive steps (see Material and Methods). For example, for a particle moving along a straight line, TA should be close to 0 degrees. At all stages, the TA distribution for somatic centromeres showed an excess near 180° (Fig. 2d), with approximately 60–65% of deflection angles above 90°, indicative of frequent backward steps and suggesting that somatic centromeres were constrained to remain within a limited domain[38]. The same trend was observed in meiocytes at leptotene and diplotene stages (61% and 65% of TA > 90°, respectively), suggesting that constraints prevented centromeres from escaping far from their initial position. In contrast, forward moves prevailed in the motion of centromeres at pachytene (53% of TA < 90°) and even more at zygotene (61% of TA < 90°), suggesting that centromeres were more prone to navigate far throughout the nuclear space at these stages.

## A. thaliana meiotic centromeres exhibit diffusive motion

To characterize the type of motion followed by centromeres, we next examined the mean square displacement (MSD), defined as the average square distance traveled during a given time interval (see Material and Methods). For a purely diffusive motion, the MSD increases linearly with time, and the slope of the curve is proportional to the diffusion coefficient; conversely, a deviation from linearity is indicative of anomalous diffusion[39,40]. Fitting a power-law function to MSD over short-time intervals (below 1 min) showed only a minor deviation from linearity (Fig. 3a, Supplementary Fig. 3). Hence, at all stages, the apparent dynamics of centromeres over short-time scales could be assimilated to a diffusion-like motion.

The estimated apparent diffusion coefficient confirmed that centromeres could travel farther in the zygotene (D = 0.045 µm²/s) and pachytene (D = 0.016 µm²/s) stages, which exhibited an -10-fold larger diffusion coefficient than the leptotene and diplotene stages (D = 0.002 µm²/s for both). Given the comparable average speed between the zygotene and pachytene stages, the slower apparent diffusion at pachytene confirmed that centromeres were more

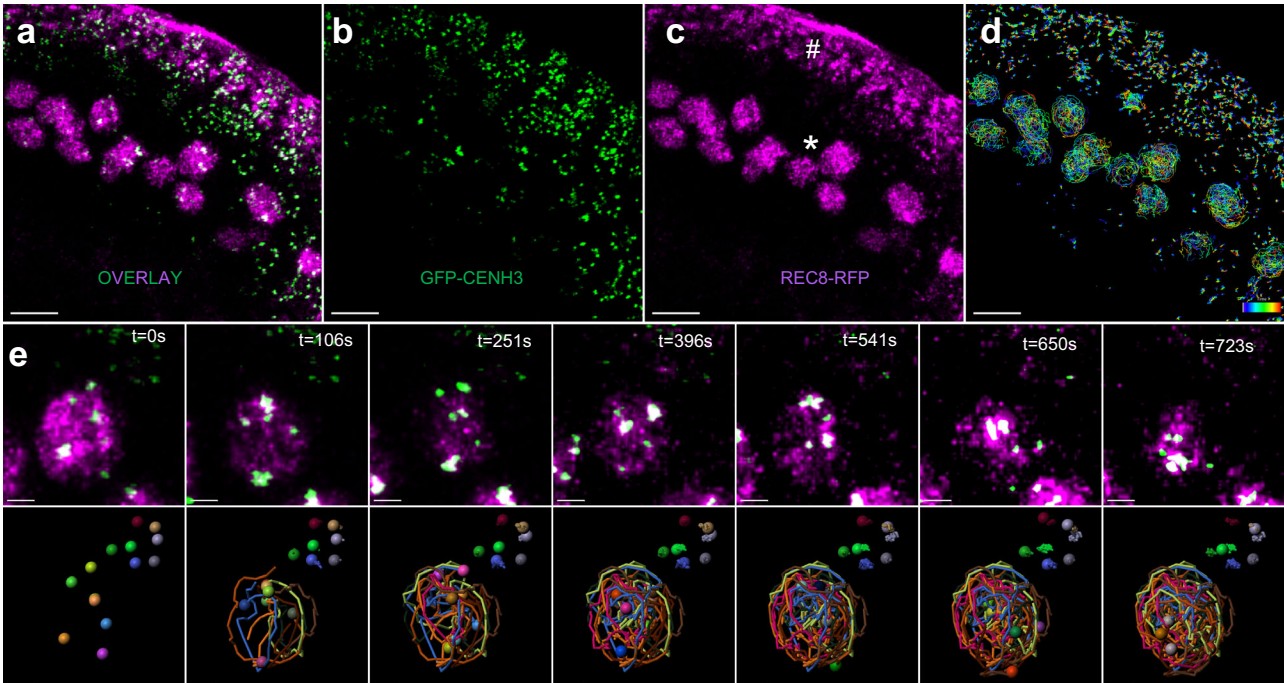

**Fig. 1 | Centromere tracking in anthers. a–c** Global views of an anther expressing GFP-CENH3 (*green*) and REC8-RFP (*magenta*). Each image represents a maximum-intensity projection of a single z-stack, corresponding to one time frame (*t* = 0). **a** Overlay of both signals. **b** GFP-CENH3 signal alone (green). **c** REC8-RFP signal alone (magenta) showing the meiocyte nuclei (marked with an asterisk). The signal observed at the edge of the anther corresponds to background fluorescence from the chloroplasts of the somatic cells surrounding the meiocytes (hashtag). **d** Reconstruction of the centromere trajectories after 12 min acquisition. The colour code corresponds to elapsed time (from 0 to 723 s). **e** Close-up view of a subset of centromeres from the above acquisition. Images are maximum-intensity projections of selected time-lapse images. The chosen area contains meiotic centromeres (colocalizing with the REC8-RFP signal) and several adjacent somatic centromeres. For each time frame, we present the overlay between the GFP-CENH3 (in green) and REC8-RFP (in magenta) (top panel) and the reconstructed trajectories of the centromeres (bottom panel). **a–d** Scale bars: 30 µm. **e** Scale bars: 1 µm. For 4D movie stack images, please refer to Supplementary Movies 1 and 2.

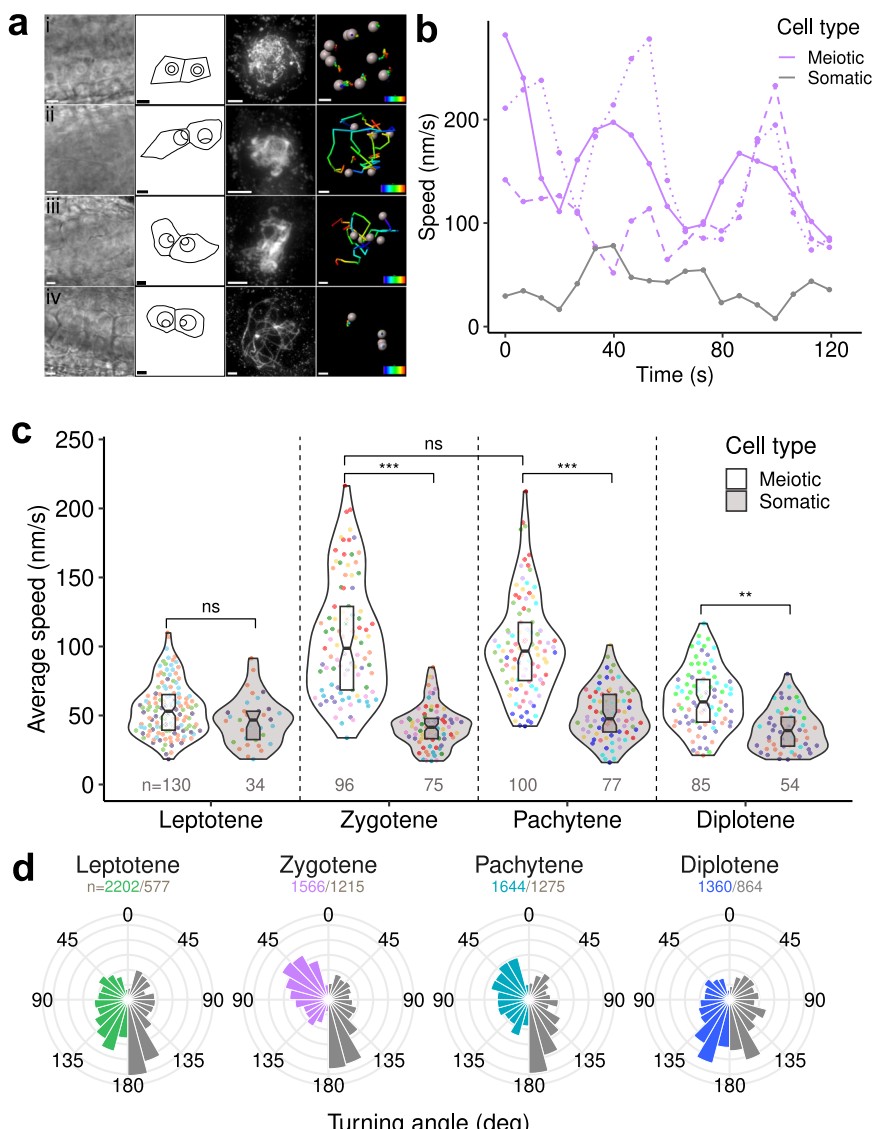

**Fig. 2 | Quantification of centromere motion in meiotic and somatic anther cells. a** Imaging and reconstruction of 3D centromere tracks in meiocytes at leptotene (i), zygotene (ii), pachytene (iii), and diplotene (iv) stages. First column: brightfield image; second column: schematic representation of the cellular outlines, nuclei and nucleoli for two meiocytes from the brightfield image; third column: DAPI staining; fourth column: one meiocyte centromere tracks represented with pseudo-colour to indicate time progression (time scale bar: 2 min). Spatial scale bars for brightfield, outlines, and DAPI: 5 μm; tracking: 1 μm. **b** Instant speed as a function of time along individual tracks at zygotene stage. Coloured curves correspond to the speed measured along three individual tracks from a same meiocyte. Different styles (dashed or dotted lines) are used to distinguish the different tracks. Grey curve corresponds to the speed of along a somatic centromere track from the same anther. **c** Distribution of average speed in meiotic and somatic anther nuclei at each meiotic stage. Boxplots indicate median values and interquartile ranges. Dot colour indicates the tracks from the same anther. Horizontal brackets show the result of pairwise comparisons (Mixed-effects model, with anther and cell as nested random factors, likelihood ratio test; $p$ values of meiotic/somatic comparisons were $p = 0.19$ (leptotene), 1.8e-06 (zygotene), 3.6e-06 (pachytene), 0.003 (diplotene); $p = 0.73$ for the comparison between meiotic average speed in zygotene and pachytene). Per-meiocyte average speed in individual anthers are shown in Supplementary Fig. 2. Boxplots extend from the first to the third quartiles of the distribution, with the middle line indicating the median value. ns: non-significative difference; **$P < 0.01$; ***$P < 0.001$. Numbers indicate numbers of analyzed tracks. **d** Distribution histograms of the instant turning angle along centromere tracks in meiotic (colour) and somatic (grey) anther cells at each meiotic stage. Histograms are normalized to unit area. Numbers correspond to the instant steps included in these analyses. Source data are provided as a Source Data file.

constrained in their displacements at this stage than at zygotene, as suggested by TA distributions. Over the four stages, the estimated diffusion coefficient in meiotic centromeres was at least ten times larger than that in somatic centromeres, where it was between $3.0 \times 10^{-4}$ and $6.2 \times 10^{-4}$ μm²/s (Supplementary Fig. 3), in accordance with previous studies[41,42]. Hence, meiotic centromeres exhibited faster apparent diffusion than interphasic centromeres at all prophase I stages, including at stages (leptotene and diplotene), where they showed only limited displacements.

Over longer time scales (10 min), the MSD in the zygotene/ pachytene stages exhibited a plateau reminiscent of confined diffusion (Fig. 3b, Black curve), which was confirmed by the close fit of the confined diffusion model equation (Fig. 3b, Magenta curve). The estimated size of the confinement domain (estimated radius = 4.01 μm) was close to the typical size of meiotic nuclei (meiotic nuclei diameter at leptotene $6.8 \pm 0.39$ μm on average, $8.8 \pm 0.53$ μm at zygotene/ pachytene and $8.7 \pm 0.26$ μm at diplotene, mean ± s.d.), suggesting that centromeres navigated at these stages within regions comparable in

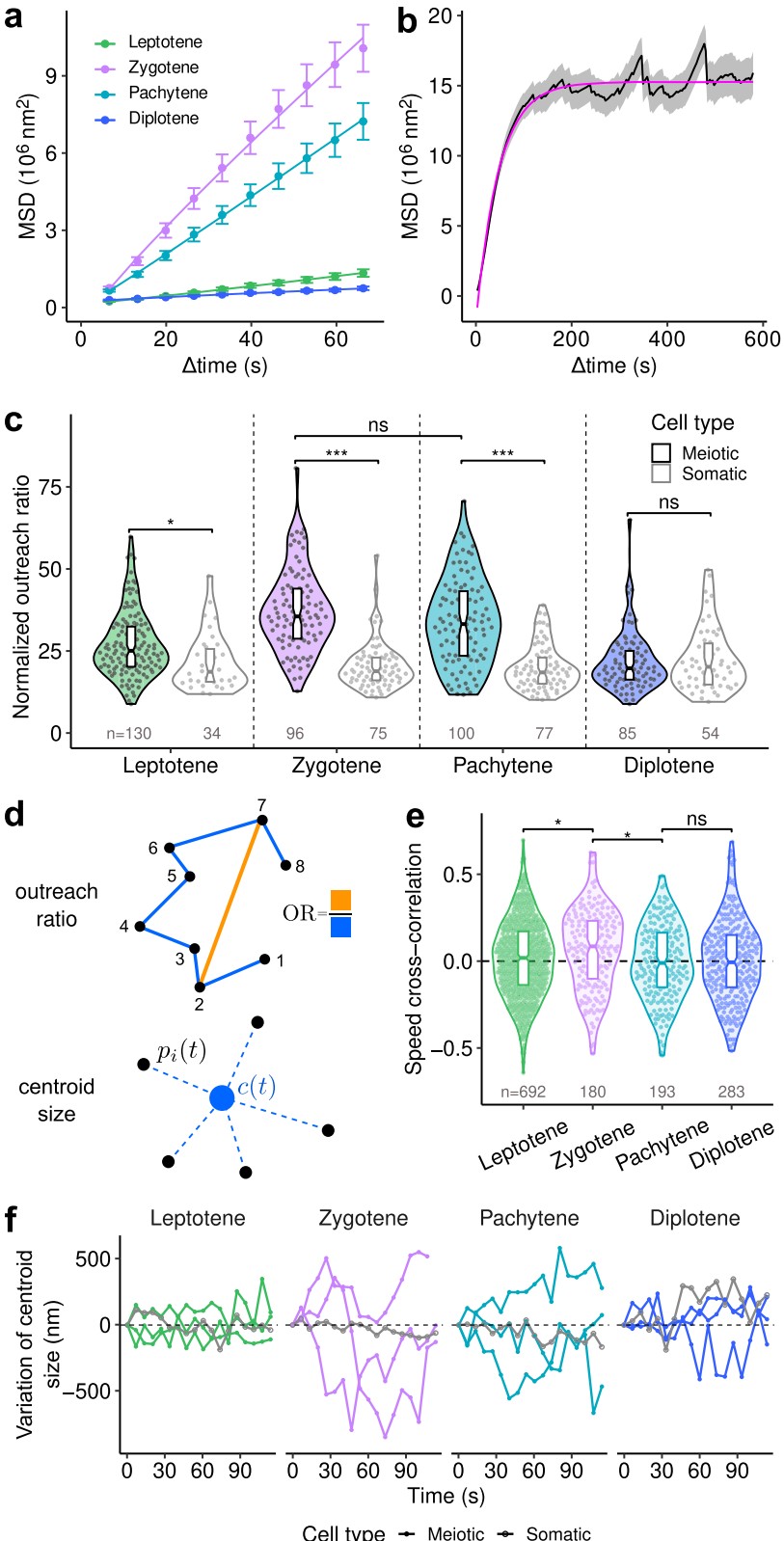

size to the nucleus. At the plateau, the MSD further showed oscillations with a period of ~150 s, which could reflect the periodicity of centromere motion through the nucleus. For example, traveling a circle of radius 4.01 μm in 150 s would require a speed of 160 nm/s, within the range of the observed average speed (Fig. 2c).

The lower diffusion coefficient during pachytene compared with zygotene led us to hypothesize that centromeres were more

constrained during pachytene. We thus examined the global shape of centromere trajectories across stages, taking the outreach ratio (OR) as a global, speed-independent descriptor of trajectory shapes. This parameter is the ratio between the maximal displacement between any two positions in the track and the total length of the track, normalized by the track duration (see Material and Methods and Fig. 3d). There were significant OR differences across stages for meiotic centromeres

**Fig. 3 | Quantification of centromere trajectories and correlations between trajectories. a** Mean square displacement (MSD) curves for the different meiosis stages computed over short tracks (total duration 2 min). Control MSD (somatic centromeres) are shown in Supplementary Fig. 3. Error bars: average ± s.e.m. Numbers of tracks: 130 (leptotene), 96 (zygotene), 99 (pachytene), 85 (diplotene). **b** Average MSD curve (*Black*) computed over long tracks (total duration 10 min; $N = 72$ tracks) at the zygotene stage. *Grey envelope*: average ± s.e.m. *Magenta*: least-square fit of the confined diffusion model. **c** Distribution of the outreach ratio in meiotic and somatic nuclei during meiosis. Numbers indicate numbers of analyzed tracks. Horizontal brackets show the result of pairwise comparisons (mixed-effects model, with anther and cell as nested random factors, likelihood ratio test; $p$ values of meiotic/somatic comparisons were $p = 0.03$ (leptotene), 1.4e-06 (zygotene), 1.6e-05 (pachytene), 0.75 (diplotene); $p = 0.44$ for the comparison between meiotic outreach ratio in zygotene and pachytene). Boxplots extend from the first to the third quartiles of the distribution, with the middle line indicating the median value. ns: non-significative difference; *$P < 0.05$; ***$P < 0.001$. **d** Parameters for describing individual and collective shapes of motion. The outreach ratio is determined by the maximal displacement over the length of the cell track segment. At time $t$, the centroid size is defined as the root mean-squared distance between the positions of centromeres [$p_i(t), 1 \leq i \leq \#centromeres$] and their average position [$c_i(t)$] at that time. **e** Distribution of speed cross-correlation between centromere tracks within meiocytes. Horizontal brackets show the result of comparisons between successive stages (mixed effect model with anther and cell as nested random factors ns: non-significant difference; $P = 0.03$ (leptotene/zygotene), 0.01 (zygotene/pachytene), 0.74 (pachytene/diplotene). Each point represents a pair of tracks from the same meiocyte. Numbers give the number of pairs at each stage. Boxplots extend from the first to the third quartiles of the distribution, with the middle line indicating the median value. *$P < 0.05$. **f** Temporal dynamics of centroid size for centromere configurations within meiocytes. Each curve displays the difference between the centroid size at any time step and the centroid size at time 0. At each stage, the four curves correspond to one somatic cell (Grey) and three meiocytes (Colour) from the same anther. Source data are provided as a Source Data file.

(mixed-effects model, $P = 0.001$) but not for somatic centromeres ($P = 0.95$). Compared with somatic nuclei, OR was much larger in meiocytes at both zygotene and pachytene stages (Fig. 3c). This shows that, independent of speed differences, centromeres were exploring a relatively larger space along their trajectories at these stages than at leptotene and diplotene stages. However, the computed difference between the average OR at zygotene and pachytene was not significant (mixed-effects model, $P = 0.36$). Overall, these results show quantitative and qualitative specific changes in the dynamics of centromeres during meiotic prophase, with increased speed and more extensive exploration of nuclear space in zygotene and pachytene stages. Intriguingly, the diffusion coefficient at pachytene was lower than that at zygotene despite similar speeds and ORs.

### *A. thaliana* centromere movements are poorly coordinated within meiocytes

We next asked whether centromere movements were coordinated. We noticed that centromeres from the same anther generally had similar average speeds (Fig. 2c, coloured dots and Supplementary Fig. 2), which was confirmed by a significant anther effect (mixed-effects model, $P < 0.001$). These similar average speeds between centromeres of the same anther revealed some degree of synchronization in the motion of centromeres across meiocytes, which could be a direct consequence of the synchronization of meiosis within an anther[43,44]. We then asked whether such coordination could occur between the centromeres of a given meiocyte. The speed cross-correlation between the different tracks of a meiocyte was generally close to zero. It barely exceeded 0.4 in absolute value (Fig. 3e), suggesting that there was no strong coupling between the instant speed of different centromeres of a given cell over the considered duration (2 min). The only exception was at the zygotene stage, where speed cross-correlation was more important compared to leptotene and pachytene (mixed-effects model, $P = 0.028$ and $P = 0.011$, respectively), suggesting that some coordination in speed across centromeres of the same cell could occur at this stage.

Finally, we examined the relative positioning of different centromeres within each cell. We computed for each meiocyte and at each time step the centroid size (the root mean-squared distance between the positions of centromeres and their average position, Fig. 3d), which quantifies the dispersion of centromeres within the nucleus[45]. Plotting the temporal evolution of the difference between centroid size and its value at the beginning of the track showed substantial variations at zygotene and pachytene stages, with alternating phases of increasing, decreasing, or stable centroid size (Fig. 3f). Hence, the relative positioning of centromeres within cells was highly dynamic, ruling out the possibility that centromeres obeyed only a shared, rigid global motion, as could result, for example, from mere nuclear rotation. Overall, although centromere movements were coordinated between meiocytes of the same anther, individual centromere motions appeared independent of each other over the duration of the analyzed tracks. However, this does not exclude the possibility of punctually synchronized motion over shorter intervals.

### Key meiotic recombination proteins are not needed for rapid centromere movements, but SUN1 and SUN2 are

We then explored the centromere dynamics in meiotic mutants affected in various aspects of the meiotic process: either in chromosomal axis structure (*asy1, asy3, rec8*), synaptonemal complex transverse filament composition (double mutant *zyp1ab*) or recombination (*spo11, dmc1, hei10, fancm*)[46–54]. We analyzed GFP-CENH3 dynamics in mutant anthers with meiocyte shapes comparable to zygotene or pachytene stages in the wild type. We compared wild-type zygotene and pachytene stages pooled together due to the inability to distinguish between the zygotene and pachytene stages in most of the mutants. We found that none of these backgrounds altered meiotic prophase centromere movement, either in average speed (Fig. 4a, b, Supplementary Fig. 4) or centromere trajectory (Fig. 4c).

Out of the five plant SUN-domain proteins described in plants[55], SUN1 and SUN2 play redundant roles in meiosis[35,56]. We therefore analyzed GFP-CENH3 dynamics in the *sun1 sun2* double mutant. Compared to wild-type, we observed a drastic diminution in the average speed of meiotic centromeres ($54 \pm 2$ nm/s. $n = 65$ tracks) compared to wild-type zygotene and pachytene speeds ($P = 0.0012$, mixed effect model with anther and cell as random factors and genotype as fixed effect; Fig. 4a, b) and a distinct distribution of turning angles compared with wild-type (Fig. 4c), with prevailing backward steps (TA > 90°). This comparison indicates that in *sun1 sun2*, centromeres exhibited frequent back-and-forth motions and tended to remain stationary. Combined with their reduced speed, this resulted in reduced apparent diffusion of *sun1 sun2* centromeres in MSD analysis (Fig. 4d; $D = 0.002$ μm²/s). Altogether, *sun1 sun2* meiotic centromeres displayed a behavior closer to that observed in wild-type somatic cells, demonstrating the key role of SUN1 and SUN2 in controlling the meiotic prophase centromere movements in *A. thaliana*.

### The nuclear envelope of meiocytes undergoes significant reorganization during zygotene and pachytene

We then investigated the dynamics of SUN1 and SUN2 during meiotic prophase I, using either living tissues expressing GFP-tagged versions of SUN1 or SUN2[57] or through immunolocalization. Both SUN1-GFP and SUN2-GFP exhibited similar expression patterns and will be collectively referred to as SUN-GFP hereafter. We combined the SUN-GFP markers with a nucleoporin (NUP54) fused to RFP (as a marker of the nuclear pores)[58] together with the GFP-CENH3 marker. At leptotene, SUN-GFP and NUP54-RFP were present on all the nuclear periphery,

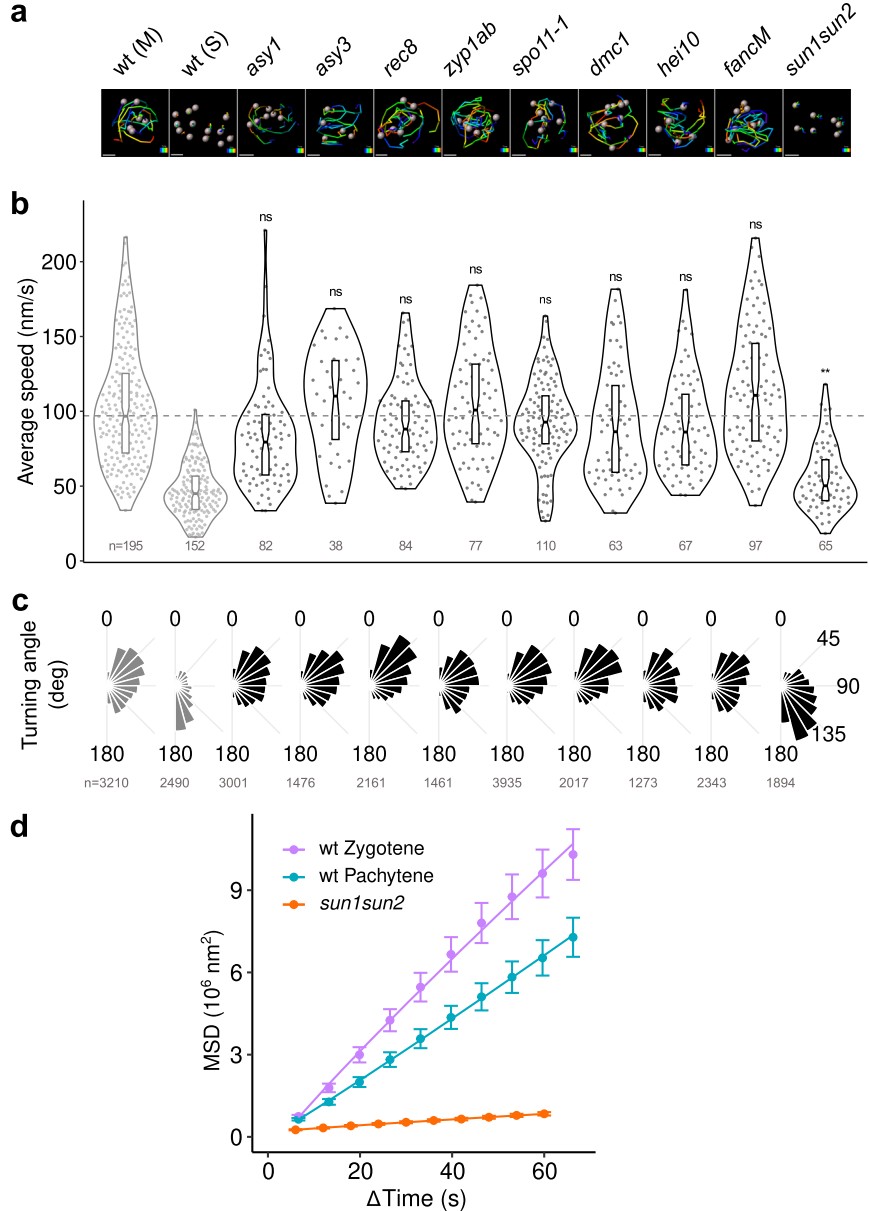

**Fig. 4 | Quantification of centromere motions in mutant meiocytes.**
**a** Reconstructed 3D tracks in wild-type meiocyte (wt, M), wild-type somatic cell (Wt, S), and mutant meiocytes (2 min acquisitions). Colour code corresponds to elapsed time. Scale bar: 2 μm. **b** Distribution of average speed in wild-type (*Grey*) and mutant (*Black*) nuclei at zygotene-pachytene stage. Numbers indicate the number of analyzed tracks. Results of pairwise comparisons between meiotic tracks in each mutant and wild-type are indicated as ns ($p < 0.05$), and ** ($p < 0.01$) (mixed-effects model, with anther and cell as nested random factors, likelihood ratio test for mutant/wt comparison: $p = 0.18$ (*asy1*), 0.92 (*asy3*), 0.43 (*rec8*), 0.81 (*zyp1ab*), 0.51 (*spo11-1*), 0.60 (*dmc1*), 0.45 (*hei10*), 0.36 (*fancM*), 0.002 (*sun1 sun2*). Boxplots extend from the first to the third quartiles of the distribution, with the middle line indicating the median value. **c** Distribution histograms of the instant turning angle along centromere tracks in wild-type and mutant meiotic nuclei at zygotene-pachytene stage. Histograms are normalized to unit area. Numbers correspond to the instant steps included in the analysis. **d** Mean square displacement (MSD) curves computed over short tracks (total duration 2 min) in wild-type nuclei at zygotene or pachytene stages and in mutant nuclei at zygotene/pachytene. Error bars: average ± s.e.m. Numbers of tracks: 96 (wt/zygotene), 99 (wt/pachytene), 65 (*sun1 sun2*). For each mutant, we analyzed between 3 and 7 anther locules and between 2 and 3 meiocytes per locule. Source data are provided as a Source Data file.

and centromeres were located near the nuclear membrane (Fig. 5a) ($n = 7/7$ anthers). In zygotene and pachytene, we observed a polarization of the SUN-GFP signal on one side of the nucleus ($n = 10/11$ anthers), confirming an observation made previously in *A. thaliana*, maize and rice[35,56,59]. During this polarization step, the nucleoporin NUP54 was polarized on the opposite side of the nucleus, globally complementing the SUN-GFP signal (Fig. 5). This complementarity between the domains of the NE where SUNs or nuclear pores localize was confirmed by immunolocalizing SUN proteins (SUN1 and SUN2) and NUP153 on 3D-preserved meiocytes

(Supplementary Fig. 5). We additionally observed that the SUN-GFP signal was often not homogeneous, forming threads at the NE surface ($n = 20/23$ anthers), which were visible only on 3D reconstructions (Fig. 5a, arrows). At diplotene, SUN-GFP and NUP54-RFP resumed their position on the whole periphery of the nucleus in a less uniform and more patchy manner than in leptotene, but with overlapping SUN-GFP and NUP54-RFP signals (Fig. 5a) ($n = 7/7$ anthers).

We took advantage of the polarized patters of SUN-GFP and NUP54-RFP in the nuclear envelope during zygotene and pachytene to explore whether the centromere RPMs we detected are associated with a global

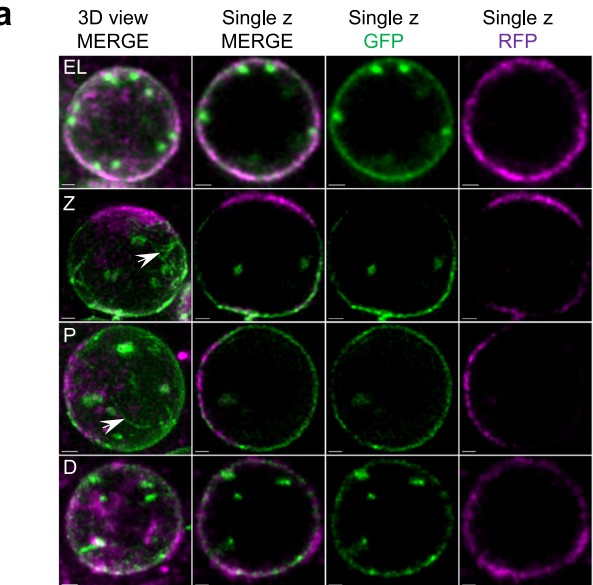

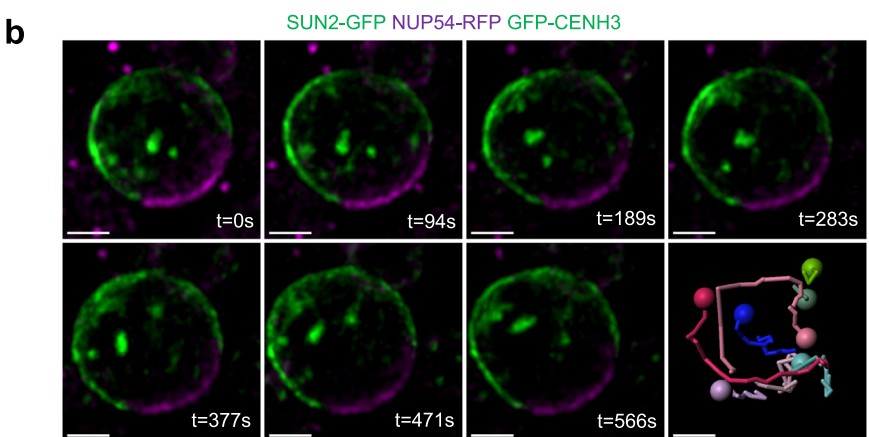

**Fig. 5 | Nuclear envelop reorganization during meiotic prophase in living anthers. a** Single z-stack acquisitions of male meiocyte nuclei expressing SUN1-GFP, NUP54-RFP, and GFP-CENH3 at different stages. Images in the left column are maximum-intensity projections of the z-stacks (3D views). Images in the other columns are single z slices. At early leptotene (EL), SUN1-GFP and NUP54-RFP signals overlap at the nuclear envelope. The centromeres (bright GFP signals) are detected close to the nuclear envelope. At zygotene (Z) and pachytene (P), the SUN1-GFP and NUP54-RFP signals occupy complementary regions of the nuclear envelope. Centromeric signals can be observed in the nucleoplasm but no longer at the nuclear periphery. At diplotene (D), SUN1-GFP and NUP54-RFP signals progressively resume their initial positions, with centromeric signals still observed in the nucleoplasm. Scale bars: 1 μm. White arrows indicate SUN1 threads.
**b** Maximum-intensity projections of a selection of time-lapse images of a meiocyte nucleus expressing SUN2-GFP, NUP54-RFP, and GFP-CENH3. For each time frame, we show the overlay between the green signals (GFP-CENH3 and SUN2-GFP) and the magenta signal (NUP54-RFP). The last panel shows the reconstructed trajectories of the centromeres, which reveal a clear displacement of the centromeres over the acquisition period. On the contrary, SUN2-GFP and NUP54-RFP signals remain in the same positions, indicating that centromere displacement is not accompanied by nuclear rotation. Refer to Supplementary Movie 7 for the 4D movie stack images. Scale bars: 2 μm.

rotation of the nucleus, as reported in *Drosophila* and mice[8,9,60]. Live acquisitions on male meiocytes expressing the three markers GFP-CENH3, NUP54-RFP, and SUN-GFP revealed that centromere movements (GFP-CENH3 signal) do not correlate with any detectable rotation of the nucleus. This was demonstrated by the lack of displacement of the NE markers during centromere movement (Fig. 5b, and Supplementary Movie 7), definitively ruling out the possibility that RPMs in *A. thaliana* could be linked to a global nuclear rotation.

These drastic changes in NE organization during meiotic prophase were confirmed by immunolocalizing SUN1 and SUN2 on 3D-preserved meiocytes together with markers of the chromosome axis (REC8) and HEI10 that allows accurate prophase staging (Fig. 6 and Supplementary Fig. 6). HEI10 is a ZMM protein[48] that shows a dual

immunolocalization pattern: from zygotene to early pachytene, it colocalizes with ZYP1, while from late pachytene to diakinesis, it accumulates at CO-designated sites as bright foci ([61] and Supplementary Fig. 6). Our observations confirmed that at early leptotene (when the nucleolus is central and centromeres are associated with the NE), the SUN proteins covered the entire nuclear membrane (n = 12/12, n = number of cells) (Fig. 6a, EL). Then, progressing from late leptotene (LL, peripheral nucleolus, and entire axis formed, n = 9) to zygotene (Z, n = 13, n = number of cells), polarization of the SUN signal was observed. 3D reconstruction of the SUN signal revealed that this polarization is corresponding to a non-homogeneous redistribution of the SUN proteins, where SUN was depleted from several areas of the NE, forming holes of different sizes in the NE (Fig. 6a, arrows). Non-

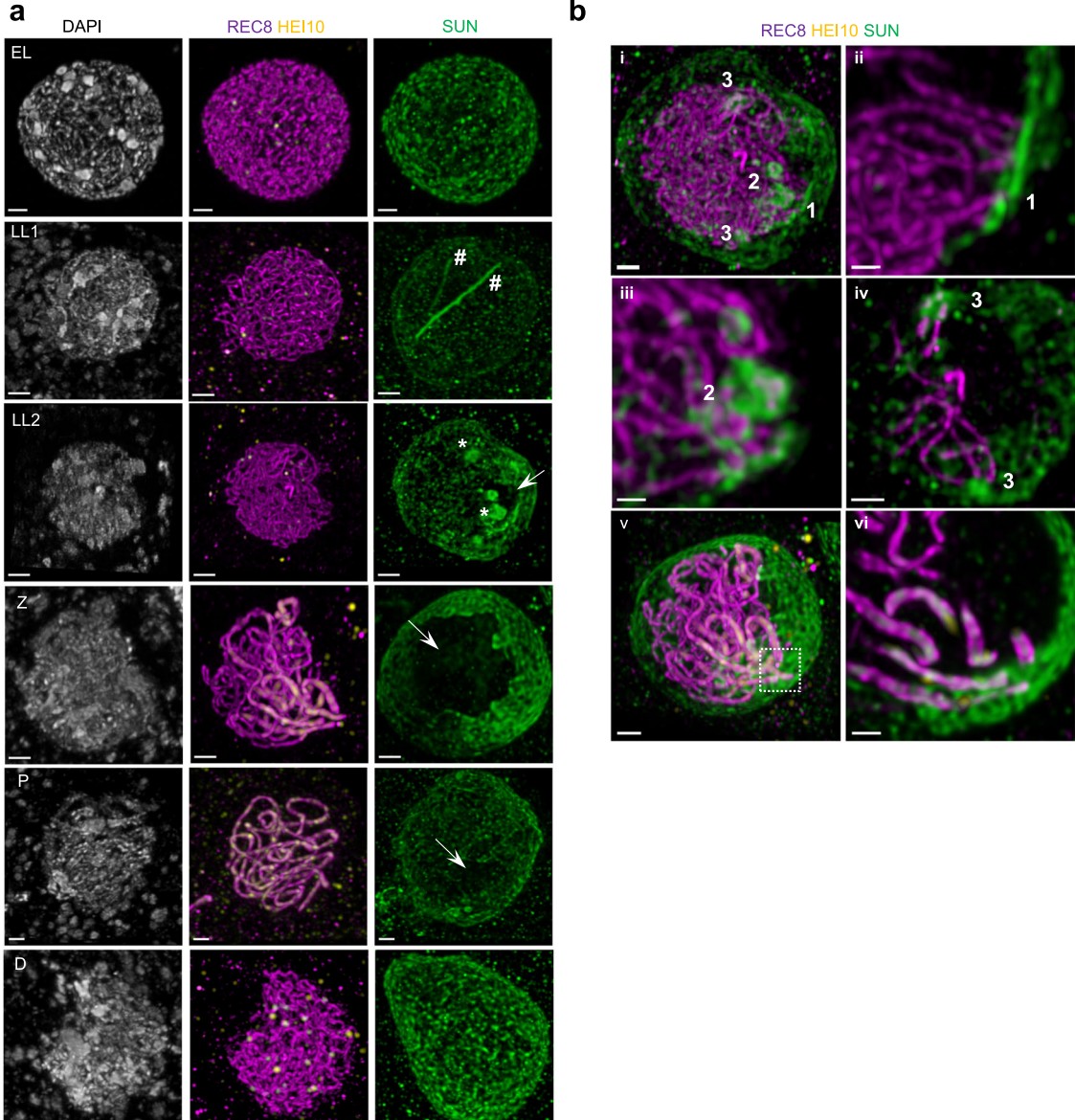

**Fig. 6 | SUN1 and SUN2 dynamics during male meiotic prophase. a** Co-immunostaining of SUN1 and SUN2 (SUN) with REC8 and HEI10 on 3D-preserved male meiocytes. The HEI10 signal is used to determine the meiocyte developmental stages: no HEI10 signal is detected in early and late leptotene (EL and LL, respectively), HEI10 signal is linear at zygotene (Z) and pachytene (P) where it loads on synapsed regions. At diplotene (D), HEI10 forms bright foci corresponding to class I COs. All images correspond to maximum-intensity projections of the whole z-stacks. SUN aggregates are indicated by hashtags (#) (stretches) or asterisks (clumps); SUN depletion by arrows. See Supplementary Fig. 6 for individual signals. Scale bar: 1 μm except for LL1 (2 μm). **b** Different views of the LL2 late leptotene cell (i–iv) and of the zygotene cell (v–vi) from **a** to illustrate SUN1 and SUN2 accumulation at regions where chromosome ends are detected. i: Maximum projection intensity of the late leptotene LL2 cell z-stack. Scale bar: 1 μm. ii–iv: Close up views on numerated areas of i (maximum projection intensity of a subset of the stack). Scale bar: 0.5 μm. v: Maximum projection intensity of the zygotene cell z-stack. Scale bar: 1 μm. vi: Close up view of a zone of the NE where chromosome extremities are detected. Scale bar: 0.5 μm. Refer to Supplementary Movie 8 (LL2 nucleus) and Supplementary Movie 9 (Z nucleus) for the 3D views of both nuclei with telomere identification.

homogeneous SUN signal was observed in 2/3 of the late leptotene cells ($n = 6/9$) and was systematic in all the cells at zygotene ($n = 13$). In pachytene, SUN redistribution was less frequent and less intense (observed in half of the meiocytes, $n = 8/16$). At diplotene, the SUN protein again covered the whole NE ($n = 5/6$) (Fig. 6a). The NE areas where there is a SUN signal when polarization occurs always cover the region where the nucleolus is located (Supplementary Figs. 5 and 6).

Immunolocalization experiments also confirmed the formation of SUN aggregates at the NE surface either as clumps (Fig. 6a, *asterisks*) or threads (Fig. 6a, *hashtags*). SUN clumps were not observed at early leptotene ($n = 0/5$, $n$ = number of cells) and rarely at diplotene ($n = 1/6$, $n$ = number of cells) but were very frequent at late leptotene ($n = 8/9$,

$n$ = number of cells), zygotene ($n = 13/13$, $n$ = number of cells) and pachytene stages ($n = 14/16$, $n$ = number of cells). Colabeling of NE (antibodies against the SUN1 and SUN2 proteins) and the chromosome axis allowed us to visualize telomere-led chromosome association with the NE. We observed that in the vast majority of the cases, SUN clumps corresponded to regions of the NE where chromosome ends are anchored (Fig. 6b, Supplementary Movies 8 and 9), suggesting that chromosome attachment to the NE requires SUN accumulation. We detected SUN threads at most prophase I stages in different proportions (late leptotene 3/9, zygotene 7/13, pachytene 6/16, and diplotene 5/6), and they did not correlate with a specific chromosomal organization or region.

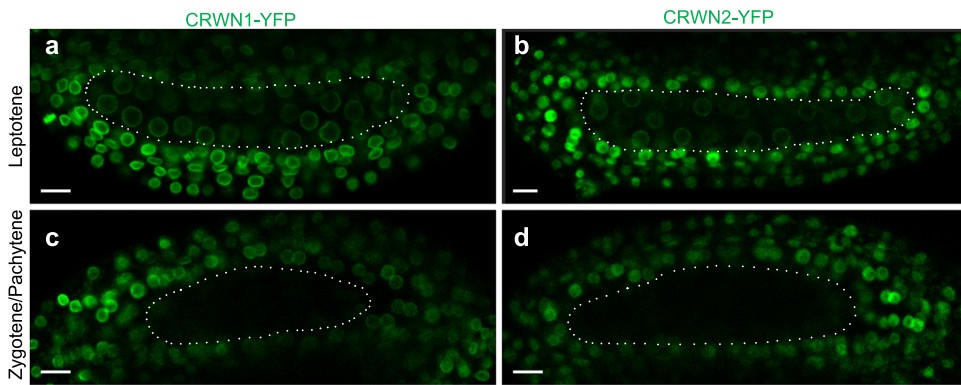

**Fig. 7 | Lamina reorganisation during meiotic prophase. a, b** Maximum-intensity projections of leptotene anthers expressing CRWN1-YFP or CRWN2-YFP (respectively). **c, d** Maximum-intensity projections of mid-prophase anthers expressing CRWN1-YFP and CRWN2-YFP (respectively). Scale bars: 10 μm. At early stage (Leptotene, **a** and **b**), CRWN1-YFP and CRWN2-YFP label the nuclear envelopes of all cells within the anthers (meiotic and somatic). In older anthers (zygotene or pachytene stages, **c** and **d**), the CRWN1-YFP and CRWN2-YFP disappear from the meiocyte nuclei. Dotted lines surround the meiocyte compartment of the anthers.

In order to deepen into meiotic NE organization, we then investigated the dynamics of two plant functional homologs of lamins, CRWN1 and CRWN2[62,63]. Lamins form a filamentous structure at the internal side of the NE[64]. We observed that at leptotene, both CRWN1-YFP and CRWN2-YFP were detected in somatic and meiotic nuclei (Fig. 7a, b). In zygotene and pachytene stages, however, we observed a complete disappearance of both CRWN1-YFP and CRWN2-YPF signals in meiocytes, while the signal was still present in the surrounding somatic cells (Fig. 7c, d). Therefore, zygotene and pachytene rapid chromosome movements occur simultaneously with a spectacular restructuring of the NE that includes the accumulation of the SUN1 and SUN2 proteins at regions containing the anchoring points of chromosome ends and the reorganization of the lamina-like layer.

### Telomere dynamics in wild-type *A. thaliana* prophase

The enrichment of SUN1 and SUN2 in the regions of the NE where chromosomes are attached led us to wonder whether and how these proteins could influence telomere distribution during meiosis. To analyze telomere dynamics during meiotic prophase I, we used chromosome axis extremities as a proxy for telomeres (see Materials and Methods, Fig. 8a, b, Supplementary Fig. 7 and Supplementary Movies 10–12). This approach allowed us to identify the majority of the 20 *A. thaliana* telomeres (on average, 19 telomeres in zygotene ($n = 14$ cells), 20 in pachytene ($n = 17$ cells) and 16 in the diplotene stage ($n = 15$ cells)). Of these identified telomeres, we determined which were positioned at the nuclear periphery and which were positioned in the nucleoplasm (Fig. 8c, e). For that, we manually delimited the nuclear periphery location (see Materials and Methods) and we analyzed the average distance to the nuclear periphery surface of the telomeres (Fig. 8c). Telomeres positioning at the nuclear periphery is interpreted as the visualization of the meiotic chromosome anchorage to the NE, as detected by colabeling of the chromosome axis and NE (Fig. 6b). We observed that for all stages analyzed (zygotene, pachytene, and diplotene), the vast majority of the telomeres were located at the nuclear periphery but that they were progressively detached of the NE from pachytene to diplotene (Fig. 8e).

Then, we analyzed telomere distribution in terms of clustering (Fig. 8d, f–h) to investigate telomere bouquet formation. For bouquet quantification, we did not consider the chromosome arms that contain the nucleolar organizing regions (NORs, four arms corresponding to the short arms of chromosomes 2 and 4). They are tightly associated with the nucleolus, and after late leptotene, they are grouped, embedded in a large domain of heterochromatin corresponding to chromosome 4 KNOB, and often located at the interface between the nucleolus and the NE[65] (Supplementary Fig. 8). This particular behavior could bring a bias in telomere clustering quantification. We defined that telomeres formed a bouquet when a minimum of 37% of the telomeres were found in a cluster (equivalent to a minimum of 6 telomeres out of 16 in a cluster). We observed telomere bouquet formation in all the zygotene cells analyzed ($n = 14$), most of the pachytene cells (90%, $n = 17$), and only 50% of the diplotene cells ($n = 15$) (Fig. 8f, Supplementary Table 1). Among the cells that display a telomere bouquet, we observed that the average number of telomeres per cluster tended to diminish from zygotene to diplotene (Fig. 8g), starting from an average of 10 telomeres per bouquet at zygotene to 8 at diplotene (Fig. 8g, h, Supplementary Table 1).

This telomere dynamics shows that, in *A. thaliana*, the telomeres form a bouquet from zygotene to pachytene that gradually disappears during diplotene, concomitantly with telomere detachment from the NE.

### SUN1 and SUN2 are essential for telomere anchorage to the nuclear envelope, and nucleolus positioning at the leptotene/zygotene transition

We then investigated telomere behavior in the *sun1 sun2* double mutant. The *sun1 sun2* double mutant does not complete full synapsis[35]. Consequently, telomeres were identified in zygotene-like stages ($n = 22$) and compared to wild-type zygotene and pachytene stages pooled together. In *sun1 sun2*, an average of 18 telomeres were detected compared to 19 in wild type for the equivalent stages (Z and P). Out of these 18 telomeres, only 40% (7.3 on average) were positioned at the nuclear periphery (Figs. 8e, 9, Supplementary Fig. 9, Supplementary Movies 13 and 14), while the rest were observed in the nucleoplasm (60%), either close to the nucleolus or more scattered in the nucleoplasm (Fig. 9). This is strikingly different from wild-type zygotene and pachytene cells, where the vast majority of the telomeres were found at the nuclear periphery (93% of the detected telomeres).

In addition, we observed that in contrast to the wild type, where the nucleolus at zygotene and pachytene was always located at the periphery of the nucleus, half of the cells in *sun1 sun2* (50%, $n = 11/22$) showed a central nucleolus (Figs. 8i and 9).

Considering telomere clustering in *sun1 sun2*, we hardly ever observed a proper bouquet (as defined as a minimum of 6 clustered telomeres) at the nuclear periphery (Fig. 8f, g), contrary to the wild type, where 95% of the zygotene and pachytene stages showed a clear bouquet. However, we observed telomere clustering in the nucleoplasm, with 90% of the cells (20/22) showing an average of 8.5 telomeres grouped together, close to the nucleolus (Fig. 9). Notably, in *sun1 sun2*, NOR-bearing chromosomes (short arms of chromosomes 2 and 4) were difficult to detect and therefore are likely part of the nucleolus-associated clusters.

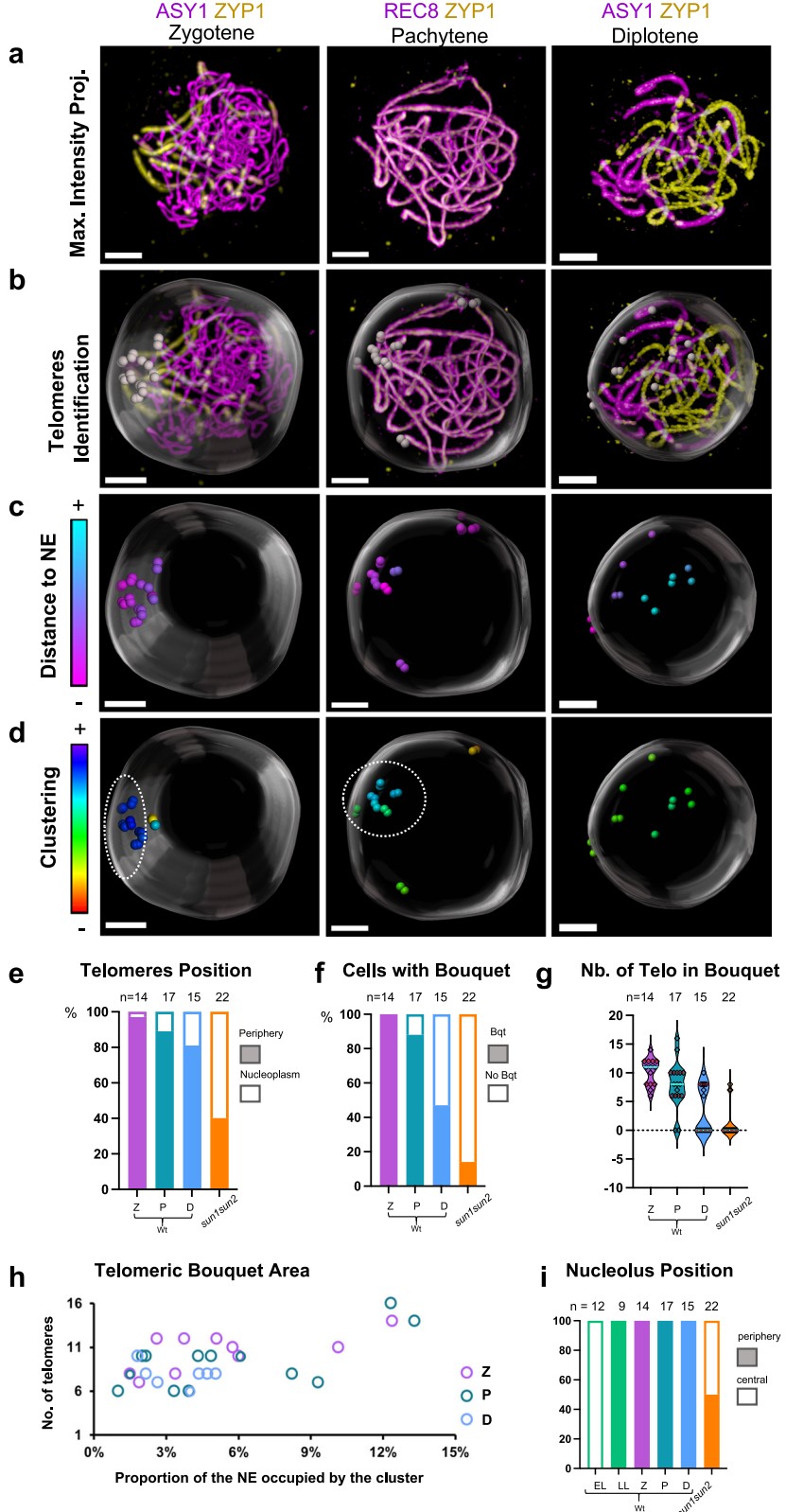

These results show that during prophase I, SUN1 and SUN2 control the positioning of telomeres in the nuclear periphery, specifically in the zygotene and pachytene stages, when rapid chromosome movements occur. Additionally, these proteins are necessary for the proper migration of the nucleolus to the nuclear periphery that occurs in wild type at the leptotene/zygotene transition.

## Discussion

### Centromeres in *Arabidopsis* present highly dynamic autonomous trajectories during the zygotene and pachytene stages

Here, we show that *A. thaliana* centromeres are subject to spectacular chromosome movements during the zygotene and pachytene stages of meiotic prophase. Contrary to what has been found in mouse, maize

**Fig. 8 | Telomeres dynamics in *A. thaliana* prophase. a** Immunostaining of male meiocytes at different developmental stages using anti-ASY1 or anti-REC8 in conjunction with anti-ZYP1. **b** Telomeres (grey spots) and nuclear periphery segmentation (transparent spheres) on nuclei from **a**. **c** Colouring of telomere spots according to their distance to the nuclear periphery surface. **d** Colouring of telomere spots based on their level of clustering. The spots corresponding to the chromosome extremities of the NOR-bearing chromosomes (chromosomes 2 and 4) were removed from the quantification. Examples of the identification of the NOR-bearing chromosomes can be seen on Supplementary Fig. 8. Doted lines indicate the clustered telomeres (bouquet). **a, b** Maximum-intensity projections. Separate channels can be viewed in Supplementary Fig. 7 and 3D view animations in Supplementary Movies 10–12. **a–d** Scale bars: 2 μm **e** Histogram illustrating the percentage of detected telomeres located at the periphery of the meiocytes (filled bars) or within the nucleoplasm (empty bars) at different developmental stages, in wild type or in *sun1 sun2* double mutant (zygotene and pachytene-like cells).

**f** Histogram showing the percentage of cells with (filled bars) or without (empty bars) a bouquet of telomeres at the nuclear periphery. A bouquet is defined here as a minimum of 6 chromosome ends (out of 16) clustered at the nuclear periphery. The chromosome ends from the short arms of chromosomes 2 and 4 are not considered for bouquet quantification because their behaviour is modified by their link with the nucleolus (NOR-containing arms). **g** Violin plot showing the number of telomeres detected within the bouquet (cells without a bouquet are indicated at zero). The violin plot indicates median values (cyan lines) and interquartile ranges (red lines). **h** Plot showing the relationship between the area occupied by the bouquet (as a percentage of the NE area) and the number of telomeres in the bouquet. Cells without a bouquet are not shown. **i** Histogram showing the proportion of cells with a peripheral (filled bars) versus an internal (empty bars) nucleolus. EL Early Leptotene, LL Late Leptotene, Z Zygotene, P Pachytene, D diplotene, Bqt bouquet. Numbers indicate the number of cells analyzed. Source data are provided as a Source Data file.

and Drosophila[9,31,60], we can exclude that these movements are due (even partially) to a global nuclear rotation since the centromere displacements happen in nuclei where the polarized nuclear pore signals (NUP54-RFP) were static (Fig. 5b, Supplementary Movie 7). We also found that the centromere displacements within each meiocyte were poorly correlated, suggesting that chromosomes move principally independently from each other (Fig. 3e, f). *A. thaliana* centromeres move fast: 100 nm/s on average, more than three times faster than chromosome migration during anaphase I (LC pers data). This average centromere speed value is in the same range as the rapid prophase movements reported in other species (400 nm/s. in maize, 110 nm/s. in mice, 40–160 nm/s. in *C. elegans*, 200–400 nm/s. in *S. cerevisiae* and 90 nm/s. in *S. pombe*)[7,9,12,29,31,66,67]. We observed that the instantaneous speed of the individual centromeres was also very variable over time, ranging from 0 to 500 nm/s. in very short-timelaps (Fig. 2b, Supplementary Fig. 1). Past studies in other species have shown that rapid prophase chromosome movements were not necessarily continuous in time, with different types of motion observed over the time course of individual trajectories. ast, In budding yeast, abrupt displacements of individuals or groups of chromosomes were described, as well as alternation between paused, directed, and straight motion of telomeres on short-time scales[11,68]. Similarly, alternated types of motion events have been described for the pairing centers in the worm[13] and mouse telomeres[9]. Although we observed significant fluctuations in the displacement speed of *A. thaliana* centromeres at zygotene (Fig. 2b), we did not find any evidence for such alternated sequences of contrasted motions. However, given that the frequency of alternation between sequences reported in other species was on the order of a few seconds, we cannot rule out the possibility that the temporal resolution of our data did not allow us to detect such transitions. Although it is challenging, analyzing data with temporal resolution on the order of the second will help assess the existence of such heterogeneities in *Arabidopsis*. Additionally, analyzing the dynamics of chromosomal regions other than centromeres will be necessary to accurately compare the nature of chromosome displacement between *Arabidopsis* and other organisms.

*A. thaliana* centromere movements at zygotene and pachytene showed almost perfectly linear mean-squared displacement curves (Fig. 3a). In other species, prophase chromosome movements result from active motion mediated from the cytoskeleton via nuclear envelope complexes[69]. For example, chromosome movements in *C. elegans* were abolished by microtubule depolymerization and reduced in dynein knock-outs[13], while in budding yeast, they were affected by actin depolymerization and impaired in Myosin2 inducible mutants[11,20]. Supralinear MSD curves are expected for motor-driven motion, as reported for subtelomeric pairing centers in *C. elegans*[13]. To resolve our paradoxical observation of linear MSD curves (indicative of diffusive motion[39]), one might hypothesize that telomere-led movements at the surface of *Arabidopsis* meiotic nuclei are themselves completely

random. However, this view is not supported by the existence of a telomere bouquet in *Arabidopsis* male meiosis (Fig. 8) and the requirement for directed motion during bouquet formation[70,71]. A more plausible interpretation is that, given that there are no telocentric chromosomes in *A. thaliana*, the nature of the motion observed at centromeres differs from that generated at distant telomeres. This differentiation is consistent with the less dynamic movement of internal chromatin marks compared to chromosome ends observed during maize meiosis[31]. Polymer simulations have also shown that diffusive motion can be observed at intermediate chromosome regions even when distal regions undergo directed motion[72]. Last, subdiffusive relative motion between homologous loci has been observed in budding yeast[73], despite active chromosome movements generated by cytoskeleton-associated motors[20].

The diffusion of centromeres was more efficient at zygotene than at pachytene (Fig. 3a). However, we did not find a significant difference between the average speed of centromeres at these two stages (Fig. 2c). This difference in diffusion suggests that constraints extrinsic to chromosomes are differentially exerted by their environment in the nuclear space between zygotene and pachytene, with a more restricted accessible space at pachytene. The more compacted chromatin state at zygotene[74] may explain why, for identical DNA content, relatively more nuclear space is accessible to moving chromosomes at this stage. Additional factors may also contribute to the difference in diffusivity, including the fact that chromosomes are fully synapsed at pachytene and therefore larger and probably also very different in terms of rigidity than unsynapsed chromosomes. In addition, although it has seldom been documented[75], an increase in nucleolar volume during meiosis may also result in greater restrictions on chromosome movements at pachytene than at zygotene.

We detected rapid prophase chromosome movements throughout zygotene and pachytene, indicating that these movements are not significantly influenced by the degree of synapsis between homologous chromosomes or the recombination progression. We confirmed this conclusion by the mutant data analyses we performed, where we tested the effect on chromosome movement of several mutations impairing either the chromosome axis (*asy1*, *asy3*, *rec8*), synaptonemal complex transverse filament (*zyp1*), or different steps of meiotic recombination (DSB formation-*spo11-1*, strand invasion-*dmc1*, ZMM pathway-*hei10*, non-CO DSB repair-*fancM*). These mutants have very variable phenotypes. *spo11-1* and *dmc1* are defective in homologous chromosome pairing and synapsis, and they do not form any CO[53,54]. In *zyp1*, *hei10*, and *fancM*, homologous chromosomes pair properly but synapse only in the latter two[48,49,51]. CO formation in *hei10* is extremely reduced[48] but considerably boosted in *fancM* (Class II CO increase)[49] and *zyp1* (Class I CO increase)[51]. Interestingly, none of these defects impact

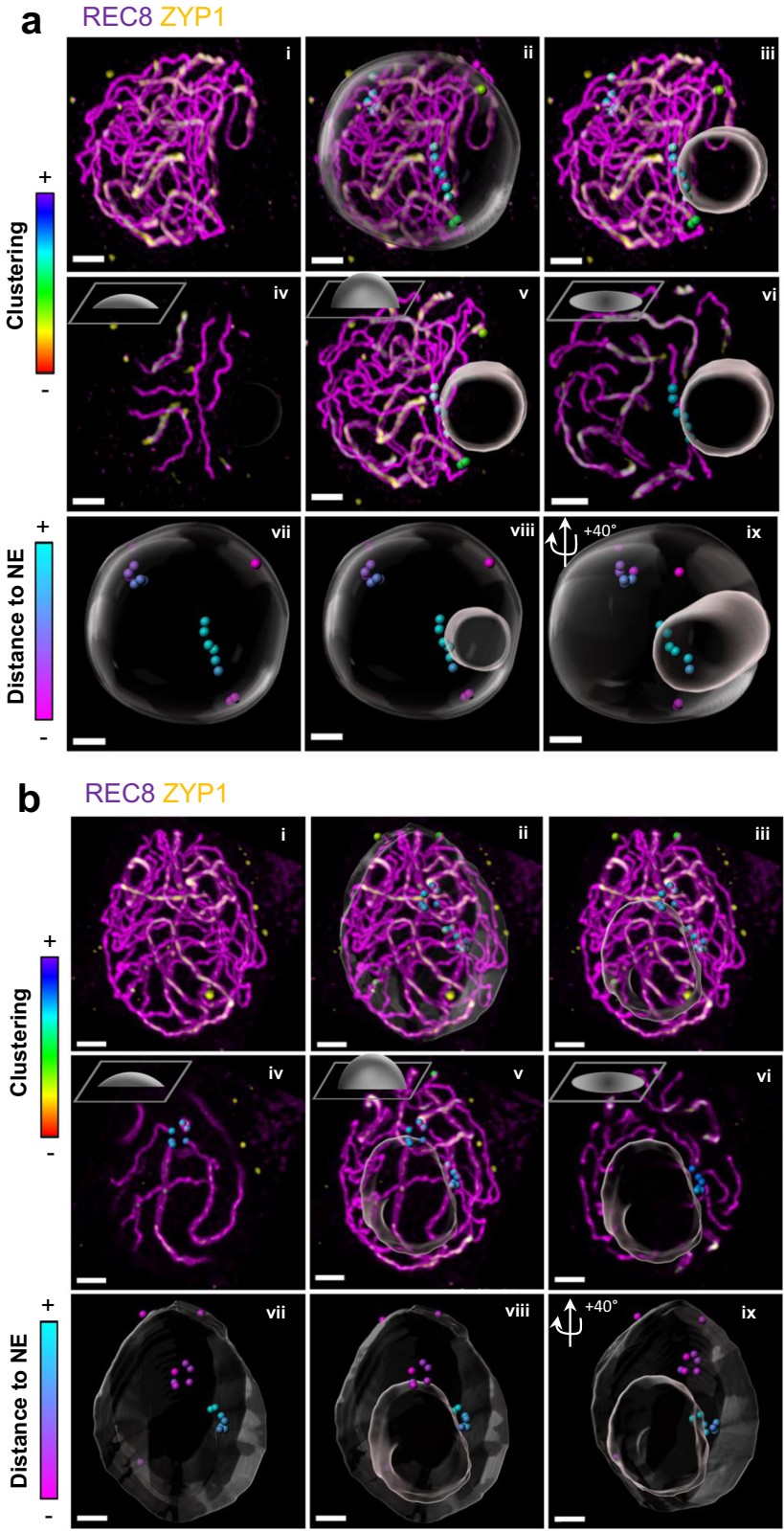

chromosome movement in *Arabidopsis*, either in terms of average speeds or global trajectories (turning angle metrics). A situation that could be different from other organisms since in mice, *C. elegans* and *S. cerevisiae* chromosome movements are moderately but consistently affected in several meiotic mutants (*Dmc1*[−/−], *Hfm1*[−/−], and *Sycp3*[−/−] in mouse)[9] (*rec8D, dmc1D* and *zip1D in Sc*)[28] (*spo11, him-3* and *htp-3* in *C. elegans*)[12].

## Rapid chromosome movements in *A. thaliana* correlate with a major reorganization of the nuclear envelope

We observed that the NE of *A. thaliana* meiocytes undergo major reorganizations during zygotene and pachytene. During these stages, we detected NE territories enriched in SUN1 and SUN2 proteins that alternate with territories where the nucleoporins accumulate (Figs. 5 and 6 and Supplementary Fig. 5). We further observed that the

**Fig. 9 | Telomere and nucleolus dynamics in *sun1 sun2* double mutant.** Immunostaining of 3D-preserved male meiocytes of *sun1 sun2* mutant against REC8 (magenta) and ZYP1 (yellow). In *sun1 sun2*, 50% of the cells exhibit a peripherical nucleolus (**a**) similar to wild type (see Supplementary Fig. 10), and the remaining 50% show an internally-located nucleolus (**b**). Telomere positions (chromosome ends) are indicated by spots (ii–ix), the location of the nuclear periphery by a large transparent sphere (ii, vii–ix), and the position of the nucleolus by a small transparent sphere (iii, v–vi, viii–ix). The colouring of the telomere spots is done either according to their clustering level (ii–vi) or to their distance to the nuclear periphery surface (vii–ix). Colour codes for telomere colouring are shown on the left of their respective panels. i–iii: Maximum-intensity projection of the z-stacks. iv–vi: Maximum-intensity projection of variable number of z slices (as depicted in the schema at the top left of each panel, iv = z 1–10, v = z 1–30, vi = z 30). vii–ix: Chromosome ends, nuclear periphery and nucleolus segmentation from different perspectives. Separate channels and 3D movie stack images are available in Supplementary Fig. 9 and Supplementary Movies 13–14. Scale bars: 2 µm.

submembrane region composed of the lamina meshwork is modified since the two lamin-like proteins CRWN1 and CRWN2 are undetectable during zygotene and pachytene (Fig. 7). The NE represents a physical barrier between the cytoplasm and the nucleoplasm. However, the LINC protein complexes and the nuclear pores connect these two cellular compartments. The LINC complexes comprise SUN proteins anchored in the inner nuclear membrane and KASH proteins anchored in the outer nuclear membrane. They are responsible for transmitting forces between the cytoplasm and the nucleoplasm and contribute to NE rigidity. Cytoskeleton fibers are physically connected to the LINC complex, and this interaction is involved in nuclear positioning and orientation, as well as nucleus shape and genome stability[76,77]. The nuclear lamina is a protein meshwork that underlies the nuclear inner membrane. It contributes to the nucleus structure and organizes the genome[64]. Studies in somatic cells showed phosphorylation's role in regulating the flexibility of the lamina meshwork[64]. Targeted depletion of the lamina in *C. elegans* oocytes provokes nuclear collapse due to rigidity loss of the nucleus, which cannot resist the forces exerted by the cytoskeleton[78]. These results suggest that the lamina is needed to counteract the forces transmitted through the LINC complex. In plants, a lamina-like structure has been observed by electron microscopy[62], and mutations in lamina-like genes cause defects in nuclear shape[79] and chromatin organization[80]. A reorganization of the lamina layer during meiotic prophase I could be a general feature. For example, in chicken male and female meiocytes at the pachytene stage, no lamin layer is detected by electron microscopy[81], but it reappears at diplotene in oocytes[82]. In *C. elegans*, remodeling of the lamina during meiosis has been described, which involves its phosphorylation[83]. In mammals, somatic lamins A, C, and B2 disappear, and they are replaced by a meiosis-specific variant of lamin A, lamin C2, which is enriched at telomere attachment sites[84]. This modification has been proposed to modify the local flexibility of the nuclear membrane. Such modifications of the resistance and/or the flexibility of the NE during meiotic prophase could explain the frequent observation of spectacular deformations of the NE (described as protrusions) that are correlated with RPM and telomere displacement at the NE[11,12,31]. These changes in NE structure and stiffness could be a prerequisite to allowing telomere attachment (accessibility) and/or displacements within the nuclear envelope.

Similarly, the reorganization of the components of the LINC complex during meiosis is present in a wide range of species. Studies in yeast, mammals, and worms have shown that the LINC complex accumulates as foci at sites where chromosome ends/pairing centers are attached[12,85–91]. Mechanisms that control the redistribution and accumulation of the meiotic SUN domain-containing proteins at the interface between chromosomes and NE during meiotic prophase only started to be understood. However, they seem to involve direct phosphorylation of SUN proteins[92–94]. This accumulation of LINC complexes at the extremities of the telomeres is likely to correspond to the "attachment plates", an electron-dense structure described in mammals that connects the synaptonemal complex with the inner nuclear membrane[95,96]. In plants, studies performed in maize, rice, and *Arabidopsis* also revealed a reorganization of the SUN domain-containing proteins in the meiotic NE during meiotic prophase, but this reorganization was far less dramatic than that in other species,

since SUN was described as polarized on one side of the NE, forming a crescent moon signal[35,56,59]. Using 3D immunolocalization, we show that the polarization corresponds to a patchy redistribution of the SUN proteins in the nuclear envelope with areas of various sizes without SUN proteins. Concomitant with this SUN redistribution, we observed an accumulation of SUN proteins as aggregates that correlate with the position of the telomeres in the NE (Fig. 6b, Supplementary Movies 8, 9). These SUN clumps could, therefore, be similar to the SUN aggregates and electron-dense attachment plates mentioned above. They have the same temporal pattern as in yeast, *C. elegans*, and mammals, since the SUN aggregates in telomeric zones in *A. thaliana* were observed during the zygotene and pachytene stages, mainly when chromosome movements are detected. We also observed that SUN proteins accumulate as long stretches. Their significance still needs clarification because they do not correlate with telomere anchoring in the NE. They could play the role of "rails" where telomere displacement occurs. Alternatively, they could correspond to SUN thread-like structures described in mouse somatic cells, which provide a mechanical link between the actin cytoskeleton and the lamin nucleoskeleton across the nuclear envelope[97].

Last, we observed a complementarity between the SUN-enriched domains in the NE and the nuclear pores, which is compatible with the observations made in insects, plants and mammals that nuclear pores are nonhomogeneously located at the NE of meiocytes and never located at the sites of chromosome attachment[98–102]. The removal of the nuclear pores from the telomere attachment area could also be linked to the global change in rigidity of the NE, since, in addition to providing a connection between the cytoplasm and the nucleoplasm, the nuclear pore scaffold provides rigidity to the NE through its cylindrical structure[103]. Collectively, these data suggest that meiotic RPMs necessitate a significant reorganization of the nuclear envelope, likely crucial for modifying its physical properties. These changes could be indispensable for facilitating telomere attachment to the nuclear envelope and their subsequent displacements.

### The SUN1 and SUN2 proteins are responsible for rapid chromosome movements, telomere attachment to the NE, bouquet formation, and nucleolus displacement

Attachment of the telomeres to the NE and their clustering in a restricted area of the NE (the bouquet) is widely conserved across species[71,84,104,105]. Even if the occurrence of the bouquet in *A. thaliana* has been sometimes questioned[35,106], 3D FISH experiments clarified that telomeres bouquet does also exist in that species[61]. Here, we developed an alternative approach to analyze telomere clustering based on the immunolocalization of chromosome axis proteins on 3D-preserved meiocytes. It presents the advantage of allowing the simultaneous detection of telomeres (chromosome ends) with any other meiotic marker (such as NE, synapsis progression, and recombination) and providing more accurate quantification of telomere clustering. We observed telomere clustering all the way from zygotene to early diplotene (Fig. 8f); however, the proportion of cells showing clustering and the intensity of clustering (number of telomeres in the cluster and cluster area) loosen gradually in pachytene and diplotene (Fig. 8f–h). Our data confirmed previous results from[35] that in the *sun1 sun2* double mutant, telomere dynamics are strongly perturbed,

and we observed that telomere association with the NE is rare in *sun1 sun2*, even if not wholly absent (Fig. 8e). We found that telomeres were more challenging to detect in the mutant background than in the wild-type background at the same stage, which can be explained by the fact that telomeres are less accessible because they are principally localized in the nucleoplasm (Fig. 8e). This shows that in *A. thaliana*, the function of SUN1 and SUN2 as the internal component of the LINC complex is conserved. The reason for residual telomere attachment in *sun1 sun2* is still unclear. One possibility is that other inner membrane-associated proteins (such as the other SUN domain-containing proteins of *A. thaliana*) partially complement SUN1 and SUN2 depletion. A situation comparable to that in mice, where there are two SUN domain-containing proteins (SUN1 and SUN2) that are meiotically expressed[107] and are only partially redundant[89,91]. In that species, the *sun1* mutant is sterile and shows a failure of telomere attachment to the NE[88], but SUN2 could be responsible for the residual telomere attachment observed in the *sun1* ko[89,91]. In *A. thaliana sun1 sun2* mutant, we observed a complete loss of centromere movement, which is in complete agreement with the known function of the LINC complex in RPMs but also suggests that the residual telomere associations with the NE that we detect in *sun1 sun2* are not robust enough to promote movement.

In most organisms, except yeast, the telomeres are located at the nucleolus and not at the NE during meiotic interphase[69], a situation probably inherited from the somatic organization of the chromosomes. In *A. thaliana* somatic cells, the telomeres are associated with the nucleolus[106,108], an association needed for proper telomere biology[109]. It is at the transition between leptotene and zygotene that telomeres detach from the nucleolus and migrate to the NE, where they become stably linked[35,61,106]. Concomitant with the appearance of telomeres at the nuclear periphery at early zygotene, we confirmed a change in the nucleolus position from the center of the cell to the periphery and observed that this always occurred in the area of the SUN-containing domains of the NE. The mechanisms controlling this transition are still a mystery. However, since the nucleolus is a structure originating from the activity of NORs, the attraction of telomeres to the NE may control the nucleolus position. In the case of terminally located NORs, such as in *A. thaliana*, the attachment of NOR-containing telomeres could physically bring the nucleolus close to the NE. This hypothesis is supported by the study of[110], which showed a clear correlation between the NOR position on the chromosomes (proximal *vs* distal) and the position of the nucleolus in spermatocytes of different mammalian species. Another possibility would be that the telomeres do not passively drag the nucleolus to the nuclear periphery. Instead, the nucleolus would migrate to the NE and bring the telomeres close to the NE where they would be remobilized. Studies investigating concomitantly nucleolus and telomere behavior at the transition between leptotene and zygotene will help to discriminate between these two hypotheses. In *sun1 sun2*, an essential proportion of the telomeres remain associated with the nucleolus, showing that SUN1 and SUN2 are needed for the telomeres to detach from the nucleolus.

SUN1 and SUN2 depletion drastically impacts chromosome dynamics during meiotic prophase. It strongly perturbs telomeres' connection to the NE, prevents bouquet formation, and completely abolishes chromosome movements. We showed that SUN1 and SUN2 are likely the inner members of the LINC complex that associates the telomeres to the NE and connects the cytoskeleton's forces to the nucleus's interior, thus controlling the rapid movements of chromosomes. It remains to be solved which are the other components of the meiotic LINC complex in plants and the proteins accompanying it, such as kinesins or telomere-associated proteins, that may participate in these same aspects as SUN1 and SUN2. In *sun1 sun2*, even if meiotic prophase movements are entirely abolished, meiotic progression is only partially perturbed[35]. The principal defect of *sun1 sun2* mutants is

a global decrease in meiotic recombination associated with partial synapsis, suggesting that in *A. thaliana*, RPMs are needed to optimize meiotic recombination efficiency. Decoupling telomere attachment from chromosome movements would be necessary to understand the exact roles of chromosome movement *vs* chromosome attachment *vs* bouquet formation.

## Methods

### Plant material and growth conditions

*A. thaliana* plants for meiosis analyses were grown in a greenhouse or growth chambers (16 h day/8 h night, 20 °C, 70% humidity). For in vitro culture, *A. thaliana* seeds were surface sterilized for 10 min in 70% ethanol + 0.05% sodium dodecyl sulfate, washed for 10 min in 70% ethanol, and grown in Petri dishes with culture medium (Gamborg B5 medium-Duchefa supplied with 0.16% bromocresol purple and 0.1% sucrose).

The *sun1-1* (SAIL_84_G10), *sun2-2* (FLAG_026E12)[35], *spo11-1-3* (SALK_146172)[111], *asy1-4* (SALK_046272)[112], *asy3-1* (SALK_143676)[50], *rec8-3* (SAIL_807_B08)[113], *zyp1-1*[51], *hei10-2* (SALK_014624)[48], *dmc1-2* (SAIL_170_F08)[114], and *fancm-9* (SALK_120621)[49] mutants were genotyped by PCR (30 cycles of 30 s. at 94 °C, 30 s. at 57 °C, and 1 min. at 72 °C) using two primer pairs (see Supplementary Table 2). The presence of fluorescent markers was detected by PCR (30 cycles of 30 s. at 94 °C, 30 s. at 56 °C, and 1 min. at 72 °C) using a primer pair with one specific to the gene and the second specific to the fluorescent protein (see Supplementary Table 2).

GFP-tagged CENH3 (GFP-CENH3) has been described in ref. 37, the RFP-tagged REC8 (REC8-RFP) line in ref. 36, GFP-tagged SUN1 (SUN1-GFP) and GFP-tagged SUN2 (SUN2-GFP) in ref. 57, YFP-tagged CRWN1 and YFP-tagged CRWN2 in ref. 65. Based on ref. 58 we chose the nucleoporin NUP54 as a marker of the nuclear pores. To generate the RFP-tagged NUP54, the *NUP54* genomic fragment was amplified with NUP54_GTWU and NUP54_GTWL primers (Supplementary Table 2). The amplification covered *NUP54* from 990 pb before ATG to the last pb before the stop codon. The PCR product was cloned and inserted into pDONR207 to create pENTR-NUP54. For plant transformation, the LR reaction was performed with the binary vector pGWB553[115]. pGWB553-NUP54 was introduced into the *Agrobacterium tumefaciens* strain C58C1. *A. thaliana* transformation was performed using the floral dip method: *Agrobacterium tumefaciens* was grown at 28–30 °C to saturation, centrifuged, and resuspended in a 1% sucrose 0.05% Silwet l-77 solution. This solution is used for dipping *A. thaliana* plants. Plants were maintained under glass overnight to increase humidity[116]. To select transformed plants, sterilized seeds were sown on culture medium with 25 µg/mL hygromycin B (Duchefa). T-DNA insertion sites for the REC8-GFP, GFP-CENH3 and NUP54-RFP lines were determined by whole-genome sequencing (see below).

### Whole-genome sequencing

DNA from the REC8-GFP, GFP-CENH3, and NUP54-RFP lines was prepared for whole-genome sequencing using the NucleoSpin Plant II kit (Macherey Nagel). DNA extraction was performed on 15-day-old in vitro culture plants. Library preparations and Hi-Seq Illumina sequencing (2 × 150 bp paired-ends) were performed by Eurofins Genomics (https://eurofinsgenomics.eu/). T-DNA insertion detection was performed with several approaches combined into a single workflow (Supplementary Fig. 11). The first used a standard pipeline for genomic variation detection (SNP and structural variation: SV). For each sample, paired-end reads were trimmed via Trimmomatic (version: 0.39; parameters: TRAILING:3)[117] and aligned onto the *A. thaliana* genome (version: TAIR10) concatenated with the T-DNA sequence using BWA mem software (version: 0.7.12; parameters: -B 4 -T 30)[118]. T-DNA coverage and sequencing depth at each genome position were obtained using the SAMtools coverage and depth programs (1.15.1). Manta was run (1.6.0)[119] to detect SV. The second approach used the

tool TDNAscan[120] to characterize the T-DNA insertions. This tool does not map reads on concatenated genomes but instead selects reads that map onto the T-DNA sequence (IR1) and aligns them onto the reference genome (IR2). Clustering IR2 provides the insertion site and a fasta of the inserted sequence; Analysis of the CIGAR code of the soft-clipped reads the orientation of the T-DNA at the insertion site (Supplementary Table 3). To visualize the alignments at the insertion sites and into the T-DNA sequences, we used the Integrative Genomics Viewer (IGV)[121]. Whole-genome Hi-Seq Illumina data were submitted to ArrayExpress (E-MTAB-13442).

### Live imaging of *A. thaliana* anthers

Live-cell imaging was performed as described in ref. 33. Flower buds of 0.4–0.6 mm were isolated on a slide. Buds were carefully dissected to isolate undamaged anthers. Anthers were transferred onto a slide topped by a spacer (Invitrogen Molecular ProbesSecure-Seal Spacer, eight wells, 9 mm diameter, 0.12 mm deep) filled with 8 μL of water and covered with a coverslip. Time lapses were acquired using a Leica TCS SP8 AOBS (Acousto-Optical Beam Splitter) confocal microscope (https://www.leica-microsystems.com/) and LAS X 3.5.0.18371 software. Images were acquired using the harmonic compound system, plan apochromatic, confocal scanning2 (HC PL APO CS2) ×20/0.75 IMM and HC PL APO CS2 ×63/1.20 WATER lenses upon illumination of the sample with an argon laser and diode-pumped solid-state laser (561 nm). GFP was excited at 488 nm and detected between 494 and 547 nm. RFP was excited at 561 nm and detected between 584 and 629 nm. Detection was performed using Leica HyD detectors. The cell shape determination and nucleolus position were made using the brightfield PMT Trans detector. Deconvolution was performed using the lightning deconvolution option. Acquisition parameters are given in Supplementary Tables 4–5.

### Meiocyte staging for live imaging acquisitions

We determined the relationship between brightfield observations, centromere mobility and meiotic stages by DAPI-staining of the anthers used for live imaging. After time-lapse acquisitions, single anthers were fixed in 30 μL of 3:1 fixative (3 vol. EtOH: 1 vol. acetic acid). Then, anthers were placed on a glass slide and washed in 5 μL of water. Water was replaced by 10 μL citrate buffer (10 mM trisodium-citrate, pH adjusted to 4.5 with HCl). Citrate buffer was removed, and anthers were digested for 5 min. at 37 °C on a heating plate with 10 μL digestion mix (3% [wt/vol] pectolyase Y-23 [MPBiomedicals-CAT#320952], 0.3% [wt/vol] Driselase [Sigma-D8037] 0.3% [wt/vol] cellulase [Onozuka R10] [Duchefa-C8001] 0.1% sodium azide in 10 mM citrate buffer). The digestion mix was removed, and anthers were dilacerated with thin needles to release meiocytes. After adding 10 μL of 60% acetic acid, the slide was incubated on a hot block at 45 °C for 1 min, and the cell suspension was stirred with a hooked needle. Another 10 μL of 60% acetic acid was added and stirred for one more minute. The cell suspension drop was surrounded by fresh 3:1 fixative, and the slide was rinsed with fixative. For DAPI staining, a drop of DAPI solution (2 μg/mL) in Vectashield (Vector Laboratories) was added. Slides were observed using a Zeiss Axio Imager Z2 microscope and Zen Blue software. Images were acquired using a Plan-Apochromat ×100/1.4 Oil M27 objective, Optovar 1.25×Tube lens. DAPI was excited at λ 335–385 nm and detected at λ between 420 and 470 nm.

### Nuclei size measurement

Each nuclei diameter was measured twice using the measurement point module in Imaris 9.7.2 software (Oxford Instruments). The average diameter was used to determine the average diameter of leptotene, zygotene/pachytene and diplotene. For leptotene, 11 nuclei were measured based on REC8-GFP and NUP54-RFP signals. For the zygotene/pachytene and diplotene stages, 33 and 7 nuclei, respectively, were measured based on SUN1-GFP and NUP54-RFP signals.

### Centromere tracking

For wild-type centromere tracking, we analyzed three meiotic nuclei and several somatic centromeres for each anther locule to quantify chromosome movements. Significant meiotic centromere displacements were observed in zygotene and pachytene, but in 7% of the acquisitions (total number of anthers, $n = 28$) meiotic centromere displacement could not be differentiated from somatic ones. These latter acquisitions were not included in the quantitative analyses. Between five and eight different anther locules from different individuals were analyzed for each stage. Centromere positions were identified at each time point, and the trajectories were determined by connecting the positions of individual centromeres Image processing and analysis were performed with Imaris 9.7.2 software (https://www.oxinst.com/) using the "Spot" module. The REC8-RFP signal was used to define the area (meiocyte) on which to perform the tracking. Spots were placed using the automated detection tool based on the GFP signal (default parameter, estimated XY diameter 1 μm) and manually corrected. Spot tracking over time was performed using the auto-regressive motion tracking algorithm with the software's default parameters. A single track was traced when two centromeres met, and the two signals became indistinguishable. When a centromeric signal splits into two individualized centromeres, the initial track is continued, and a second track starts from the splitting time point. It should be noted that the different centromeres of *A. thaliana* cannot be discriminated, which implies that when spots split, a track might switch from one chromosome to another.

### Centromere movement quantification

Using the instantaneous centromere displacements (dx, dy, dz) exported from Imaris, we reconstructed the complete individual trajectories (x, y, z positions of each centromere as a function of time). Trajectories whose duration was less than 15 s. were considered too short for meaningful analysis and were excluded from the dataset. Trajectories located along the boundary of the image domain were also excluded from further analysis to minimize potential artifacts due to boundary effects. Finally, a sample drift correction was systematically applied by removing from all centromere trajectories in the same acquisition the average trajectory of the somatic centromeres tracked in that acquisition.

Several measures were calculated on each track (Supplementary Table 6). Descriptors related to instant displacements included the instant speed, its average along the track, and the instant turning angle. Descriptors related to the whole tracks included the total displacement TD (sum of instant displacement lengths over the track) and the maximal distance MD (maximum distance between any two positions along the track). The OR, defined as the ratio between MD and TD, was computed to quantify the exploration of the nuclear space by centromeres and normalized by multiplying with the square root of track duration[122]. To characterize the motion type, we calculated the MSD, defined as the average squared displacement of centromeres at all possible time intervals along tracks[39,40]. We considered different diffusion models depending on the trajectory length. On short trajectories, we fitted a non-linear power-law model $MSD(t) \propto t^{\alpha}$ and a Brownian motion model (MSD linearly increasing with time with a slope proportional to diffusion coefficient). On long trajectories, we fitted the confined diffusion model $MSD(t) \approx R^2 \left\{ 1 - a \exp\left( -\frac{bDt}{R^2} \right) \right\}$, which accounts for spatial constraints that restrict the centromere movements. To assess the synchronization between the centromeres of a given meiocyte, we computed the cross-correlation of instant speed between pairs of tracks using Kendall's tau. We also quantified the positions of centromeres relative to each other at a given time step by the centroid size, the root mean-squared distance between each centromere and the average position of the centromeres at the considered time step[45]. The dynamics of the relative positions between

centromeres along a track were characterized by plotting the difference between the centroid size at any time step and its value at the beginning of the track.

We used mixed-effects models to assess the significance of the observed differences in the calculated parameters between the different stages. These models take into account the nested structure of our data, where multiple trajectories were observed within each cell and multiple cells were observed within each anther. Prophase stage and genotype were considered as fixed effects and anther and cell were treated as random effects, except when testing for an anther effect.

All quantifications were performed under the R software (R Core Team 2021) using in-house scripts. The OR was computed using CelltrackR[123]. Statistical analyses were performed under the R software. The lme4 package[124] was used for mixed effect models.

### Immunofluorescence studies

Immunolocalization was performed on 3D-preserved meiocytes as described in ref. 61 with the following modifications: 5 to 10 flower buds were used for a single gel pad, buds were digested for 20 min at 37 °C in 0.3% (w/v) Pectolyase Y-23 (MP Biomedicals), 0.3% (w/v) Driselase (Sigma) 0.3% (w/v) Cellulase Onozuka R10 (Duchefa). The gel pads were mounted in Vectashield (Vectorlabs), primary antibodies were incubated for 2–3 nights, washes after primary and secondary antibodies were performed in PBS 1× + Triton 0.1%, 4 times 30 min, and secondary antibodies were diluted 1:250. Antibodies were diluted in PBS 1× + BSA 3% + Tween 0.2%. The list of the primary antibodies used can be found in Supplementary Table 7. Secondary antibodies were conjugated with Alexa 488, Alexa 568, or Alexa 647 (Thermo Fisher). Observations were made using a Leica TCS SP8 AOBS (Acousto-Optical Beam Splitter) confocal microscope (https://www.leica-microsystems.com/) and LAS X 3.5.0.18371 software. Images were acquired as described in ref. 125 but using an HC PL APO CS2 ×63/1.4 NA immersion objective lens. The fluorescent signals were recorded using the Lightning mode of LAS X software. Z-stacks with 0.13 μm intervals were acquired and deconvolved using Lightning default parameters and the adaptative-vectashield option.

### Staging of immunostained meiocytes

The 3D-preserved male meiocytes were DAPI-stained and immunolabeled for different markers: REC8, ZYP1, ASY1, MLH1, or HEI10 (Supplementary Table 7). The level of synapsis was determined based on the level of ZYP1 polymerisation and/or ASY1 depletion. Discrimination between zygotene and diplotene can be achieved based on the appearance of the DAPI and ZYP1 staining or based on MLH1 or HEI10 patterns [61] (Supplementary Figs. 6 and 7).

### Image analysis on immunostained meiocytes

Telomeres, nucleolus, and nuclear periphery delimitation, were determined using Imaris 9.7.2 software (Bitplane). To identify telomere positions, we labeled the meiotic chromosomes with antibodies against REC8 and/or ASY1 and/or ZYP1. The confocal imaging of these cells provided the resolution necessary to identify and map chromosome extremities thereby enabling the determination of telomere location. Telomeres were then tagged using the Imaris spot module in manual edition mode with the object-object statistics option selected. For the clustering quantification, the statistics code of "average distance to the 9 nearest neighbors" was used. The colourmap spectrum values were fixed in the range between 0 and 8 μm (corresponding to the average meiocyte diameter). A bouquet was defined as a telomere cluster of at least 6 telomeres at a distance range less than the median distance value (4 μm). The nucleolus and the nuclear periphery were delimited manually using the surface tool in the manual edition mode, with the object-object statistics option selected. The drawing circle mode was used with a radius between 2 and 2.5 μm for the nucleolus

and between 4.5 and 5.5 μm for the NE with 10 vertices in both. Either the SUN or the DAPI signals were used to determine the nuclear periphery position. When the SUN signal was not available, we drew the smallest sphere possible that closely surrounds the entire nuclear content based on the DAPI signal. The surface occupied by a telomere cluster was delimited manually with IMARIS, using the manual drawing distance mode taking. The proportion of the nuclear periphery occupied by a telomere cluster was measured and reported to the total area of the NE surface. The nucleolus position was identified as the region of the nucleus deprived of any chromosomal signal, as shown in Supplementary Figs. 5 and 6. The Spot/Surface statistics were then used to analyze and visualize the telomere's distance to the nuclear periphery.

### Statistics and reproducibility

Statistics analyses are indicated in the above sections. No statistical method was used to predetermine the sample size. The experiments were not randomized. The investigators were not blinded to allocation during experiments and outcome assessment. The experiments have been repeated from two to ten times.

### Reporting summary

Further information on research design is available in the Nature Portfolio Reporting Summary linked to this article.

## Data availability

The raw sequencing data generated in that study have been deposited in the EMBL-EBI ArrayExpress database under accession code E-MTAB-13442. The bioinformatic workflow used to detect T-DNA insertion sites and main outputs are available in the recherche.data.gouv repository [https://doi.org/10.57745/5Q21AR]. Source data for Figs. 2–4, 8 and Supplementary Figs. 1–4 are provided with this paper.

## Code availability

All the R software scripts that have been written to perform the quantitative analyses of wild-type and mutant centromere tracks are freely available under the GNU GPL License version 3 or above. They can be retrieved from the Data INRAE repository at: https://doi.org/10.57745/V1NNFI. The codes for T-DNA detection are available at https://github.com/ijpb-bioinformatics/msgenova.git.

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

## Acknowledgements

We wish to thank Chloé Girard, Christine Mézard, Eric Jenczewski, Rajeev Kumar, and Raphaël Mercier for critical reading of the manuscript. We thank Nadia Bessoltane for her help in whole-genome sequencing analyses and Audrey Hulot for inputs on mixed model analyses. We also thank Yuki Sakamoto and Sachihiro Matsunaga for sharing the CRWN1-YFP and CRWN2-YFP lines, Yoshihisa Oda for the SUN1-GFP and SUN2-GFP lines, and Arp Schnittger for the REC8-RFP line. This work has benefited from the support of the Institut Jean-Pierre Bourgin's Plant Observatory technological platforms PO-Plants and PO-Cyto. This research was funded by ANR (COPATT ANR-20-CE12-0006 M.G., M.T., and MeioMove ANR-21-CE12-0042, M.G., P.A., L.C., S.L.). The Institute Jean-Pierre Bourgin benefits from the support of Saclay Plant Sciences (ANR-17-EUR-0007). We are grateful to the genotoul bioinformatics platform Toulouse Occitanie (Bioinfo Genotoul, https://doi.org/10.15454/1.5572369328961167E12) for providing help and/or computing and/or storage resources.

## Author contributions

L.C., M.T.A., and S.L. performed the experimental studies, carried out the analyses and contributed to the writing of the manuscript. A.C., A.H., M.B., and J.G. performed the experimental studies. I.L.M. generated the NUP54-RFP line. D.C. and G.A. developed the whole-genome sequencing pipeline. P.A. and M.G. supervised the work, analyzed the data and contributed to the writing of the manuscript.

## Competing interests

The authors declare no competing interests.
