## [Peer Review File · Nature Communications]

REVIEWER COMMENTS

Reviewer #1 (Remarks to the Author):

In their study, Cromer et al. present a comprehensive characterization of chromosome movements in *Arabidopsis thaliana* meiocytes, providing the first exploration of this phenomenon. Utilizing live-cell imaging techniques, the authors conducted an analysis of statistical mechanics parameters such as average speed, turning angle, Mean Squared Displacement, outreach ratio, and centroid size to understand the behaviour of centromeres in subsequent stages of meiosis. The authors found significant differences between leptotene/diplotene and zygotene/pachytene showing that the centromere movements depend on the meiosis progress. Interestingly, centromere movements remain unaffected in mutants of key meiotic genes related to homolog pairing, synapsis and recombination, indicating that behaviour of chromosomes in meiosis is independent from chromosome structure. Of the mutants tested, the only clear difference was observed for mutants of SUN envelope proteins, in which chromosome movements were drastically reduced. The research delves into a detailed examination of SUN1 and SUN2 proteins during meiotic prophase in both wild-type and mutant background. Additionally, the authors pioneer an analysis of nuclear envelope dynamics, establishing correlations between LINC complex and nucleoporins data. This holistic approach provides new insights into the interplay of chromosomal movements and the nuclear envelope, highlighting the roles of SUN proteins in these processes.

My main reservation about the work concerns the part relating to the analysis of telomere behaviour during meiosis and bouquet formation. While I consider the presented results to be important and interesting, I am not convinced that the analyzed number of cells allows us to draw all the conclusions presented in the work. For some of the data presented in Figure 8, the number of cells examined for a given stage of meiosis is very small, which raises doubts as to the significance of the observed changes. This is particularly important in the context of the high variability observed between individual cells.

I also have mixed feelings about the definition of telomere bouquet proposed in this paper. Traditionally, the bouquet is formed when all chromosome ends cluster in a distinct region of the genome (see for example Zickler and Kleckner, 2016). Cromer et al. defined the bouquet when a minimum of 6 out of the 20 telomeres of *Arabidopsis thaliana* are grouped (they use the term "cluster" in their definition, but later in this context they just call it "bouquet"). Given that *A. thaliana* forms 5 bivalents, the criteria imply a bouquet formation even when 33.3% of the telomeres are not clustering, assuming fully synapsed chromosomes (and thus, only 10 visible telomeres). Moreover, as depicted in Fig. 8F, some phases show cells with no bouquet formation at all. Based on the presented data, *A. thaliana* does not exhibit a classical bouquet during meiosis. In order not to mislead readers, I would suggest using a different name, e.g. "telomere cluster" or "quasi-bouquet".

The paper includes four supplementary videos, with the longest recording spanning a maximum of 700 seconds (11 minutes). Referring to Prusicki et al., 2019, the duration of late leptotene,

zygotene, pachytene, and diplotene/diakinesis is reported to be 1.5 hours, 6 hours, 9.5 hours, and 3 hours, respectively. Therefore, I would like to ask to what extent the recording period is representative of the meiotic stages lasting hours. It would be informative to explain the cumulative time recorded for each stage. I propose providing a comprehensive summary table in the supplementary data.

Despite these reservations, I believe that this is a very valuable paper that significantly increases our knowledge of the dynamics of meiotic chromosomes in Arabidopsis and will be of great interest to the meiotic society.

Minor points:

1. The text of the manuscript could benefit from some smoothing, and the figure captions should be more detailed.
2. The nomenclature of genetic constructs used in the work is incorrect. By default, the "::" character is used to show the promoter vs. gene/CDS relationship. For example, the phrase "REC8::RFP" used in this work suggests that a construct was used in which RFP protein is expressed from the REC8 promoter. The authors should use the expression "REC8-RFP", which means that the REC8 protein was used in a translational fusion with RFP protein (attached at the C-terminus). Similar corrections should be made for all other constructs, including those in the figures.
3. I would suggest refraining from providing deviations of the centromere speed in part "A. thaliana RPMs start at the zygotene stage and cease at diplotene." Too many details make the text difficult to read.
4. Figures. I recommend including the results of the statistical tests both in the main text and in the figure captions for consistency. For example, fig.2C p-values are indicated in the text but not in the figure caption meanwhile fig.4b p-values are indicated in figure but not in text). This will enhance readability and maintain uniformity in presenting statistical information across the paper.
5. Figure 2A. Please, consider including the diagram of cell structures adjacent to the brightfield images captured by the microscope, not on top of them. This would facilitate unbiased analysis of the brightfield pictures and allows for a more comprehensive representation of the cells.
6. Figure 2D (and 4C). What is the area of the plot? does it correspond to the number of tracks or cells? does the numbers above the plots correspond to the instant steps for meiocytes or somatic + meiocytes? If only for meiocytes, what are the numbers of somatic steps?
7. Figure 3B. Please include an explanation of the representation of the purple and grey lines, despite the mention in the text of the results. This will provide clarity to readers directly interpreting the figure.
8. Figure 4B – please add also average speed for somatic cells for comparison.
9. Figure 4D. As for sun1 sun2 the authors show MSD for zygotene-pachytene, I would present wild type in the same manner, for consistency.

10. Figure 9. Please, include identification of the proteins that are immunostained in the images next to the figures.

11. Video supplementary data 1B. Regarding the computational representation, the changing colors of the centromeres (represented by balls) might be causing confusion. If the variation in colors is attributed to a change in turning angle, please explicitly state this in the legend. Otherwise, for enhanced clarity and ease of following the movements, it would be beneficial to maintain a consistent colour for each centromere throughout the video.

12. Figure Supplementary 16. Legend is missing. If this figure encompasses all the data for the recorded videos, it would be valuable to include information such as the total time of each video, the specific phase of meiosis studied in each video, and an explanation of the notation used (such as "D2, E1, F3, H0...") for clarity. Providing this information will enhance the interpretability of the figure.

13. Discussion, section 2 last paragraph. There is a small typo in: "Collectively, these data demonstrate that meiotic RPMs necessitate a significant reorganization of the nuclear envelope, likely crucial for modifying the its physical properties".

14. Materials and methods. Images analyses. In this section is stated that "Confocal imaging of these cells provided the resolution needed to identify and follow most of the chromosome axes and, consequently, to identify and map chromosome extremities". To enhance clarity, it would be beneficial to specify the percentage or proportion of chromosome axes that were successfully identified and followed.

15. Materials and methods. Meiocyte staging for live imaging acquisitions. Please, provide reference/Catalogue number for the enzymes employed.

16. Materials and methods. Centromere tracking. When is mention that "Significant centromere displacements were observed in zygotene and pachytene, but 7% of the acquisitions (n=28 anthers) showed no centromere displacement and were not included in the quantitative analyses" it would be helpful to provide clarification on what is considered as 'no displacement' and the specific criteria used for this determination. Additionally, it would be beneficial to specify whether the 'n=28 anthers' refers to the total number of anthers analyzed or only to the 7% that were discarded due to the absence of centromere displacement."

17. Materials and methods. Centromere movement quantification. In this section, is indicated that "All quantifications were performed under the R software (R Core Team 2021) using in-house scripts". For the benefit of the scientific community and to enhance transparency, it is recommended to consider uploading the in-house scripts onto a platform such as GitHub. This would allow other researchers to access and potentially utilize the scripts, fostering collaboration and reproducibility in the field.

18. Bibliography 42. Strange characters appear in the citation.

19. Bibliography 48 and 75. They refer to the same paper.

Reviewer #2 (Remarks to the Author):

The recognition and association of homologous chromosomes during meiotic prophase I is facilitated by rapid prophase chromosome movements (RPMs) and telomere clustering. However, mechanisms of RPMs in different organisms vary, and little is known about the RPM in plants. Here, Cromer et al. used a combination of live cell imaging and quantitative image analyses to investigate the movement pattern of centromeres and telomeres, as well as the correlation of these movement with nuclear envelope (NE) in *Arabidopsis* male meiocytes. They found that RPMs occur in zygotene and pachytene, but not in leptotene or diplotene. Centromeres movement pattern remain normal though lacking of key meiotic proteins such as chromosome axis, SC and recombination proteins. However, the movement is abolished in *sun1 sun2*. They also investigated the NE organization, the telomere clustering and attachment to NE and nucleolus displacement. Overall, this study provides new perspectives on the regulation mechanisms of RPMs, telomere clustering and telomere-NE attachment in plants.

I believe this work is of great importance to our understanding of rapid prophase chromosome movements (RPMs). These are both novel and interesting discoveries that will be of great interest to the wider meiosis community. However, there are some major and minor issues with the paper that the authors should address:

Major points:

1. In this research, the movement of centromeres and telomeres are studied, but the movement of the other parts of chromosomes are not studied. Does the centromere movement represent the whole chromosome movement? Perhaps other parts of chromosomes show different moving pattern from centromeres. For example, telomeres form the bouquet, but centromeres don't. It would be better to not use centromere movement to represent the chromosome movement. Therefore, I suggested to modify the title, abstract and introduction to make it more rigorous.

2. Page 5, second paragraph: "We noticed that centromeres from the same anther generally had similar average speeds (Fig. 2C, colored dots)"

If I understand correctly, the dots with same color are from the same anther. However, I don't think they have similar speeds. For example, in zygotene, the average speed of pink dots ranges from 0.1 - 0.22 $\mu\text{m/s}$, yellow dots range from 0.06-0.18 $\mu\text{m/s}$ and green dots range from 0.06-0.17 $\mu\text{m/s}$, which shows significant differences, However, the speeds of blue and purple dots look more consistent.

3. Page 6, the last sentence of the second paragraph: "Altogether, *sun1 sun2* meiotic centromeres displayed a behavior closer to that observed in wild-type somatic cells".

It would be better to add the wild-type somatic cell data to Fig 4, so that it'll be more straightforward to compare sun1 sun2 with wild-type somatic cells, and then say they displayed similar behavior.

4. Page 6, the third paragraph: "We combined the SUN::GFP markers with a nucleoporin (NUP54) fused to RFP and to the GFP::CENH3 marker."

It might be unclear for the readers why you used nucleoporin (NUP54). Could you explain the function of nucleoporin (NUP54) and purpose of using this nucleoporin marker in the experiment?

5. Page 6, at the end of the third paragraph: "Live acquisitions on male meiocytes expressing the three markers GFP::CENH3, NUP54::RFP, and SUN::GFP revealed that centromere movements (GFP::CENH3 signal) do not correlate with any detectable displacement of the NE markers (Fig. 5B, and SupData_7)"

When I look at Fig 5 and SupData_7, I found it's difficult to recognize the 5 centromere signals because SUN and CENH3 are both in GFP. When the centromeres move close to NE, it is difficult to distinguish them from each other. The lack of REC8 signal makes it uncertain if the CENH3 signal is indeed on chromosomes or just background. I suggest doing immunolocalization with REC8/SUN/CENH3 and REC8/NUP153/CENH3 antibodies, to have a better view of CENH3 and SUN and better support the conclusion that centromere movements do not correlate with any detectable displacement of the NE markers.

6. In Fig 6B: there are two other items that need to be improved

1) As shown in Figure 6Bi, in zygotene, it's difficult to recognize all the chromosome ends by just looking at the pictures, which affects the accuracy of telomere number quantification. As it has been proposed that synapsis is typically initiated in chromosome end regions in some plant species, if it's also the case in Arabidopsis, I suggest using early loading of ZYP1 or HEI10 as markers for synapsis initiation sites, which would correspond to the chromosome end/telomere regions. You could choose the cells at early zygotene which just start to synapse and have a little bit loading of ZYP1 or HEI10 on the chromosome ends, those sites can be recognized as synapsis initiation sites and telomere regions.

2) The HEI10 signal is so weak, could you make it brighter, and put separate HEI10 signal so that the synapsis initiation sites can be recognized more easily?

Minor points:

1. Page 3, “Average speed measurements confirmed the absence of specific movement at the leptotene stage, as we did not observe a difference between meiotic (55 ± 20 nm/s, $n=130$ tracks) and somatic (46 ± 17 nm/s, $n=34$) nuclei (Fig. 2C, SupData_4). This contrasted with the significant difference observed at the subsequent zygotene and pachytene stages, where the average speed of 106 ± 42 nm/s ($n=96$) and 100 ± 36 nm/s ($n=96$) exhibited a nearly twofold increase.....”

All the speed units here are nm/s, but in Fig 2 and subsequent paragraphs, the speed units are $\mu\text{m/s}$, could you change all the speed units to $\mu\text{m/s}$, to be consistent in the whole article?

2. Page 4, second paragraph: “indicative of frequent backward steps that suggested that somatic centromeres were constrained to remain within a limited domain”

Please correct the overuse of “that”

3. Page 7, the second paragraph: “Interestingly, we observed that in the vast majority of the cases, SUN clumps correspond to regions of the NE where chromosome ends are anchored (Fig.6B, SupData_8)”

How do you know these are chromosome ends? It is quite difficult to identify them in Fig 6B1. Do you have probes or antibodies for chromosome ends?

4. Page 7, the last sentence: “When we analyzed the data separating early zygotene cells (EZ, less than 50% synapsis) from late zygotene (LZ, more than 50% synapsis) and early diplotene (ED, less than 50% desynapsis) from LD (more than 50% desynapsis) (SupData_9B)”

Please change LD to late diplotene and add LD in the bracket.

5. Fig 8,

A. In the picture I only see Z, P and D, please spell out zygotene, pachytene and diplotene in the legend or add them in the picture.

B. How do you know where the boundary of NE (the transparent membrane in Fig 8B) is? Was the transparent membrane created based on some NE signal (such as SUN)?

C. In the text you defined telomere clustering as minimum 6 telomeres grouped. Could you make it clear which telomeres are grouped in one cluster in Fig 8C if this can indeed be discerned? Could you also mark the clustered telomeres in the picture?

In C and D, if they are same cells, why in the middle picture of panel C, there are 4 telomeres on the top right of the cell, but in the middle picture of panel D, there are only 2 telomeres on the top right of the cell? The left picture of panel D also shows one telomere fewer than the left picture of panel C. Could the authors explain this?

6. Page 8, the first paragraph, the last sentence: “starting from an average of 11 telomeres per cluster at the EZ to 8 at late diplotene (Fig. 8 G, H)”

1) In Fig 8G, at late diplotene, number of telomeres in bouquet in four cells are zero, only one cell has 8 telomeres in one bouquet, the average should be $8/5=1.6$, not 8. Please correct in the text.

2) Please change EZ to early zygotene.

7. Page 14, in the last paragraph:

“After adding 10 μ L of 60% acetic acid, the slide was incubated on a hot block at 45 °C for 1 min., and the cell suspension was stirred with a hooked needle.”

Please delete the period after min.

8. page 16, In the “ Immunofluorescence studies” “buds were digested for 20 min at 37 °C after fixation in 0.3% (w/v) Pectolyase Y-23 (MP Biomedicals), 0.3% (w/v) Driselase (Sigma) 0.3% (w/v) Cellulase Onozuka R10 (Duchefa).”

Confusing expression. It would be better to say: “after fixation in ...(components of fixation buffer), buds were digested for 20 min at 37 °C in digestion buffer [0.3% (w/v) Pectolyase Y-23 (MP Biomedicals), 0.3% (w/v) Driselase (Sigma) 0.3% (w/v) Cellulase Onozuka R10 (Duchefa)]”.

“The gel pads were mounted in Vectashield (Vectorlabs), primary antibodies were incubated for 2 to 3 nights, washes after primary and secondary antibodies were performed in...”

Please list the components of blocking buffer for primary antibody incubation.

9. Fig 2

B. What’s the meaning of dotted line and dashed line? Please explain in the legend. The same thing should be explained in SupData_3

D. What does the radius of the sector mean? Does it mean the proportion of the cells in this angle? Please explain in the legend.

10. Fig 3B

What's the meaning of the gray and purple line? Please explain in the legend.

11. Fig 5B,

In the figure legend, "For each time frame, we show the overlay between the green signals (GFP::CENH3 and SUN2::GFP) and the magenta signal (REC8::RFP)."

I do not see the REC8 magenta signal in the picture.

12. Fig 9A and B

There are transparent round spheres outside of the chromosome. How do you know this is the position of nucleolus? Is there any marker/antibody to detect nucleolus?

13. SupData_1:

In the video you show, the REC8 signal faded away quickly, and you also mention this in the article. How do you recognize meiotic stages without REC8 signal? Based on DAPI or cell shape?

14. SupData_10A

1) It seems the zoomed picture is not exactly the part in the square dashed frame in row 1. Could you mark it clear which part in row 1 the zoomed picture comes from?

2) On the right, there are two completely identical representation diagrams, please delete one.

15. SupData_11

1) The first column: Please don't use abbreviation of the stages, use full name of the stages. If abbreviations are used, please explain them in the legend.

2) The header of the third column: "Clusters" should correct to "Clusters/Number of cells"

3) The fourth column: What is the meaning of numerator (11, 10, 9, 8, 8) and denominator (16)? Please explain it clearly in the header.

16. SupData_12

Top picture of B: It is a merge of REC8 and ZYP1, but the ZYP1 signal looks very weak, I can hardly see it. Could you increase the intensity of ZYP1 signal in the first and the third picture of B?

Please find below, in blue, our answers to the reviewers' comments.

REVIEWER COMMENTS

Reviewer #1 (Remarks to the Author):

In their study, Cromer et al. present a comprehensive characterization of chromosome movements in *Arabidopsis thaliana* meiocytes, providing the first exploration of this phenomenon. Utilizing live-cell imaging techniques, the authors conducted an analysis of statistical mechanics parameters such as average speed, turning angle, Mean Squared Displacement, outreach ratio, and centroid size to understand the behaviour of centromeres in subsequent stages of meiosis. The authors found significant differences between leptotene/diplotene and zygotene/pachytene showing that the centromere movements depend on the meiosis progress. Interestingly, centromere movements remain unaffected in mutants of key meiotic genes related to homolog pairing, synapsis and recombination, indicating that behaviour of chromosomes in meiosis is independent from chromosome structure. Of the mutants tested, the only clear difference was observed for mutants of SUN envelope proteins, in which chromosome movements were drastically reduced. The research delves into a detailed examination of SUN1 and SUN2 proteins during meiotic prophase in both wild-type and mutant background. Additionally, the authors pioneer an analysis of nuclear envelope dynamics, establishing correlations between LINC complex and nucleoporins data. This holistic approach provides new insights into the interplay of chromosomal movements and the nuclear envelope, highlighting the roles of SUN proteins in these processes.

My main reservation about the work concerns the part relating to the analysis of telomere behaviour during meiosis and bouquet formation. While I consider the presented results to be important and interesting, I am not convinced that the analyzed number of cells allows us to draw all the conclusions presented in the work. For some of the data presented in Figure 8, the number of cells examined for a given stage of meiosis is very small, which raises doubts as to the significance of the observed changes. This is particularly important in the context of the high variability observed between individual cells.

We fully understand the concern raised by Reviewer 1, and we agree that for the data presented Figure 8, the separation into too many substages (early from late zygotene/early from late diplotene) weakened the conclusions because we ended up with too few cells in each category. We have addressed this concern by either increasing the number of cells analyzed (Figure 8i) or by considering zygotene cells as a whole and diplotene cells as a whole (Figure 8 f, g). We have adjusted the text accordingly and paid attention not to overstate our conclusions.

I also have mixed feelings about the definition of telomere bouquet proposed in this paper. Traditionally, the bouquet is formed when all chromosome ends cluster in a distinct region of the genome (see for example Zickler and Kleckner, 2016). Cromer et al. defined the bouquet when a minimum of 6 out of the 20 telomeres of *Arabidopsis thaliana* are grouped (they use the term "cluster" in their definition, but later in this context they just call it "bouquet"). Given that *A. thaliana* forms 5 bivalents, the criteria imply a bouquet formation even when 33.3% of the telomeres are not clustering, assuming fully synapsed chromosomes (and thus, only 10 visible telomeres). Moreover, as depicted in Fig. 8F, some phases show cells with no bouquet formation at all. Based on the presented data, *A. thaliana* does not exhibit a classical bouquet during meiosis. In order not to mislead readers, I would suggest using a different name, e.g. "telomere cluster" or "quasi-bouquet".

We fully understand the reviewer's concern. However, we would like to point out that upon reading the literature on the bouquet, we found out that its definition varies significantly among authors. This trend is well illustrated in the review of Rubin et al. 2020 (doi: 10.3390/cells9030696). There is a clear consensus that the term "bouquet" describes the attachment and the clustering of the meiotic telomeres in a limited region of the inner nuclear envelope. However, when it comes to the intensity of the clustering (regarding the number of telomeres that compose the bouquet or the size of the area where the telomeres cluster), a wide range of configurations have been reported. For example, in some species, the bouquet contains only one of the two chromosome ends, as discussed in Scherthan et al. (2001) who defines the bouquet as "the positioning of one or both ends of the thread-like meiotic prophase chromosomes at a limited sector of the nuclear periphery" (doi: 10.1038/35085086).

Also, it is important to note that precise quantification of the bouquet is quite rare in the literature. In many papers mentioning a bouquet, a single cell is shown with nice telomere clustering in a limited area of the nuclear envelope, but it's difficult to determine if this picture reflects a general occurrence or instead corresponds only to "the best/nicest" image. In our study, we define the bouquet as a minimum of 6 telomeres (out of 16, excluding the 4 telomeres from the NOR-bearing chromosomes in our quantifications), which indeed could be considered as a low level of clustering. However, the average number of telomeres in a bouquet is found to be 10 in zygotene, accounting for more than 60% of the telomeres clustered, on average. We also detected a significant number of zygotene cells with 12-14 telomeres clustered (corresponding to 75-90% of the telomeres), which fits with the tightest bouquet configuration of other species.

In summary, we would like to argue that the term "bouquet" encompasses variable situations. This heterogeneity could be attributed to multiple factors, such as species variability and/or temporal variability. However, what we think is important is that,

regardless of the species considered or the intensity of the clustering, the phenomenon of telomere clustering in a limited area of the nuclear envelope is conserved. Therefore, we believe it is important to retain the term "bouquet" in our study.

The paper includes four supplementary videos, with the longest recording spanning a maximum of 700 seconds (11 minutes). Referring to Prusicki et al., 2019, the duration of late leptotene, zygotene, pachytene, and diplotene/diakinesis is reported to be 1.5 hours, 6 hours, 9.5 hours, and 3 hours, respectively.

Therefore, I would like to ask to what extent the recording period is representative of the meiotic stages lasting hours.

The Reviewer is right but as often with imaging, it's all a question of compromise. Today, with the available imaging techniques, we cannot perform continuous imaging of our samples during hours with the level of resolution required for particle tracking. We deliberately made the choice to sacrifice the duration of acquisitions for the benefit of the resolution. Here we provide the quantitative data representing more than one hour and a half of acquisitions (see revised SupData_16, now Supplementary Tables 4 and 5). But these 1h30 acquisitions represent only a small percentage of the total acquisitions we made on the CENH3-GFP line. We estimate that more than 22 hours of acquisitions were made of centromere movement in wild-type anthers at the zygotene-pachytene stage without observing any deviation from what we present here. That is why, we are quite confident that the selection of acquisitions we present in this manuscript cannot be biased.

It would be informative to explain the cumulative time recorded for each stage. I propose providing a comprehensive summary table in the supplementary data. Absolutely, we have added this information in the modified SupData_16 (Now Supplementary Tables 4 and 5).

Despite these reservations, I believe that this is a very valuable paper that significantly increases our knowledge of the dynamics of meiotic chromosomes in Arabidopsis and will be of great interest to the meiotic society.

Minor points:

1. The text of the manuscript could benefit from some smoothing, and the figure captions should be more detailed.

We have edited our manuscript with the objectives of smoothing and clarifying as much as we could the text. We also modified the figure legends, providing more details to help the reader. We hope we now provide a version much more reader-friendly. Thanks very much to Reviewer 1 for his/her constructive remarks.

2. The nomenclature of genetic constructs used in the work is incorrect. By default, the "::" character is used to show the promoter vs. gene/CDS relationship. For example, the phrase "REC8::RFP" used in this work suggests that a construct was used in which RFP protein is expressed from the REC8 promoter. The authors should use the expression "REC8-RFP", which means that the REC8 protein was used in a translational fusion with RFP protein (attached at the C-terminus). Similar corrections should be made for all other constructs, including those in the figures.

Changes have been done accordingly.

3. I would suggest refraining from providing deviations of the centromere speed in part "A. thaliana RPMs start at the zygotene stage and cease at diplotene." Too many details make the text difficult to read

We agree with the reviewer that adding standard-deviations and sample sizes makes the reading less smooth than with average values alone. However, it has now become such a standard to report variability statistics associated with average estimates that we don't think it is possible to remove this information.

4. Figures. I recommend including the results of the statistical tests both in the main text and in the figure captions for consistency. For example, fig.2C p-values are indicated in the text but not in the figure caption meanwhile fig.4b p-values are indicated in figure but not in text). This will enhance readability and maintain uniformity in presenting statistical information across the paper.

We have corrected these inconsistencies. Significance levels are now mentioned in figure legends. We have also added the P-value of the mixed effect model test of the genotype effect in the comparison of average speed between wild-type and *sun1sun2* tracks.

5. Figure 2A. Please, consider including the diagram of cell structures adjacent to the brightfield images captured by the microscope, not on top of them. This would facilitate unbiased analysis of the brightfield pictures and allows for a more comprehensive representation of the cells.

We have revised Figure 2a accordingly.

6. Figure 2D (and 4C). What is the area of the plot? does it correspond to the number of tracks or cells? does the numbers above the plots correspond to the instant steps for meiocytes or somatic + meiocytes? If only for meiocytes, what are the numbers of somatic steps?

Thank you for bringing this to our attention. The numbers for somatic measurements were indeed missing. This has been corrected in the updated version of Figure 2c, which

now includes above the plots the numbers of instant steps for both somatic and meiotic cells, color-coded to match the graphs.

In Figures 2d and 4c, each histogram shows the normalized distribution of the instant turning angles over all the tracks, where the total area is equal to one. We have now added this information in the figure legends.

7. Figure 3B. Please include an explanation of the representation of the purple and grey lines, despite the mention in the text of the results. This will provide clarity to readers directly interpreting the figure.

Thank you for pointing to the missing information in Figure 3 caption. We have updated the legend to include the following clarification: “Average MSD curve (Black) computed over long tracks (total duration 10 minutes; N=72 tracks) at the zygotene stage. Grey envelope: average \pm s.e.m.. Magenta: least-square fit of the confined diffusion model.”

8. Figure 4B – please add also average speed for somatic cells for comparison.

Average speed has been added for wild-type somatic cells in Figure 4b. For the sake of consistency within the different panels of this figure, we have also added a corresponding view of tracked centromeres in Fig. 4a and the corresponding histogram of turning angles in Fig. 4c. A new “Supplementary Fig. 4” figure has been added to show average speeds in meiocytes and somatic cells in all mutants.

9. Figure 4D. As for *sun1 sun2* the authors show MSD for zygotene-pachytene, I would present wild type in the same manner, for consistency.

Thank you for your suggestion regarding the presentation of wild-type data akin *sun1 sun2*. However, we were concerned that such a presentation might compromise the consistency with figure 3a. Also, it appeared to us that readers can easily interpolate between zygotene and pachytene, if needed. Therefore, we choose to maintain the distinction between the zygotene and pachytene stages of the wild-type data in Fig.4d.

10. Figure 9. Please, include identification of the proteins that are immunostained in the images next to the figures.

The proteins which are immunodetected are now indicated.

11. Video supplementary data 1B. Regarding the computational representation, the changing colors of the centromeres (represented by balls) might be causing confusion. If the variation in colors is attributed to a change in turning angle, please explicitly state this in the legend. Otherwise, for enhanced clarity and ease of following the movements, it would be beneficial to maintain a consistent colour for each centromere throughout the video.

Absolutely, this was a mistake from us. Thanks for spotting it. We have changed the video (now Supplementary Movie 2).

12. Figure Supplementary 16. Legend is missing. If this figure encompasses all the data for the recorded videos, it would be valuable to include information such as the total time of each video, the specific phase of meiosis studied in each video, and an explanation of the notation used (such as "D2, E1, F3, H0...") for clarity. Providing this information will enhance the interpretability of the figure.

We have modified SupData_16 (Now Supplementary Tables 4 and 5) as suggested by the Reviewers 1 and 2.

13. Discussion, section 2 last paragraph. There is a small typo in: "Collectively, these data demonstrate that meiotic RPMs necessitate a significant reorganization of the nuclear envelope, likely crucial for modifying the its physical properties".

The typo has been corrected.

14. Materials and methods. Images analyses. In this section is stated that "Confocal imaging of these cells provided the resolution needed to identify and follow most of the chromosome axes and, consequently, to identify and map chromosome extremities". To enhance clarity, it would be beneficial to specify the percentage or proportion of chromosome axes that were successfully identified and followed.

We understand that our sentence was misleading because it gave the impression that the axis reconstruction was necessary for chromosome end identification, which is not the case. We changed it to "The confocal imaging of these cells provided the resolution necessary to identify and map chromosome extremities, thereby enabling the determination of telomere location."

15. Materials and methods. Meiocyte staging for live imaging acquisitions. Please, provide reference/Catalogue number for the enzymes employed.]

References and Catalogue numbers have been added.

16. Materials and methods. Centromere tracking. When is mention that "Significant centromere displacements were observed in zygotene and pachytene, but 7% of the acquisitions (n=28 anthers) showed no centromere displacement and were not included in the quantitative analyses" it would be helpful to provide clarification on what is considered as 'no displacement' and the specific criteria used for this determination.] Additionally, it would be beneficial to specify whether the 'n=28 anthers' refers to the total number of anthers analyzed or only to the 7% that were discarded due to the absence of centromere displacement."

In that sentence, “no displacement” means no detectable difference in motion between meiotic and somatic centromeres. These acquisitions, which correspond to deteriorated samples, haven’t been included in the quantitative analyses. We have clarified that in the Material and Methods section. N=28 corresponds to the total number of anthers, which is now specified in the revised text.

17. Materials and methods. Centromere movement quantification. In this section, is indicated that “All quantifications were performed under the R software (R Core Team 2021) using in-house scripts”. For the benefit of the scientific community and to enhance transparency, it is recommended to consider uploading the in-house scripts onto a platform such as GitHub. This would allow other researchers to access and potentially utilize the scripts, fostering collaboration and reproducibility in the field.

Thank you for your suggestion regarding the transparency and reproducibility of our research. We fully acknowledge the importance of sharing our scripts to foster collaboration and reproducibility in our field. Following your recommendation, we have uploaded the scripts on the French institutional repository recherche.data.gouv.fr, where they are available as open source under the GNU GPL License: <https://doi.org/10.57745/V1NNFI>.

18. Bibliography 42. Strange characters appear in the citation.
This has been corrected.

19. Bibliography 48 and 75. They refer to the same paper.
This has been corrected.

Reviewer #2 (Remarks to the Author):

The recognition and association of homologous chromosomes during meiotic prophase I is facilitated by rapid prophase chromosome movements (RPMs) and telomere clustering. However, mechanisms of RPMs in different organisms vary, and little is known about the RPM in plants. Here, Cromer et al. used a combination of live cell imaging and quantitative image analyses to investigate the movement pattern of centromeres and telomeres, as well as the correlation of these movement with nuclear envelope (NE) in Arabidopsis male meiocytes. They found that RPMs occur in zygotene and pachytene, but not in leptotene or diplotene. Centromeres movement pattern remain normal though lacking of key meiotic proteins such as chromosome axis, SC and recombination proteins. However, the movement is abolished in sun1 sun2. They also investigated the NE organization, the telomere clustering and attachment to NE and nucleolus displacement. Overall, this study provides new perspectives on the regulation mechanisms of RPMs, telomere clustering and telomere-NE attachment in plants.

I believe this work is of great importance to our understanding of rapid prophase chromosome movements (RPMs). These are both novel and interesting discoveries that will be of great interest to the wider meiosis community. However, there are some major and minor issues with the paper that the authors should address:

Major points:

1. In this research, the movement of centromeres and telomeres are studied, but the movement of the other parts of chromosomes are not studied. Does the centromere movement represent the whole chromosome movement? Perhaps other parts of chromosomes show different moving pattern from centromeres. For example, telomeres form the bouquet, but centromeres don't. It would be better to not use centromere movement to represent the chromosome movement. Therefore, I suggested to modify the title, abstract and introduction to make it more rigorous.

Absolutely, we perfectly agree with the Reviewer that we cannot extrapolate to other chromosomal regions what we observe and quantify for centromeres. We have accordingly changed the text to avoid any unjustified generalization. Nevertheless, we believe that it is consistent to consider the centromere movements we describe here as one manifestation of the global phenomenon, known as RPMs (Rapid Prophase chromosome Movements), as defined in other species. That is why we decided to keep this appellation in the title of the manuscript, and in the situations where we refer to the global phenomenon and not only to the centromeric data.

2. Page 5, second paragraph: "We noticed that centromeres from the same anther generally had similar average speeds (Fig. 2C, colored dots)"

If I understand correctly, the dots with same color are from the same anther. However, I don't think they have similar speeds. For example, in zygotene, the average speed of pink dots ranges from 0.1 -0.22 $\mu\text{m/s}$, yellow dots range from 0.06-0.18 $\mu\text{m/s}$ and green dots range from 0.06-0.17 $\mu\text{m/s}$, which shows significant differences, However, the speeds of blue and purple dots look more consistent.

Our conclusion is not only based on Figure 2c but also on the test of an anther effect, which has a strong significance as mentioned in the second part of the cited sentence: "...similar average speeds (Fig. 2c, colored dots), which was confirmed by a significant anther effect (mixed-effects model, $P < 0.001$)". This test implies that part of the variability observed in Fig. 2c is explained by differences between anthers, or stated the other way round, by more homogeneous speed within individual anthers than between different anthers. We have forgotten in our first version to refer here to Supplementary Fig. 2, which shows per-meocyte average speed in individual anthers. Except possibly for leptotene, variations within anthers are generally smaller than overall variations. The

reference to Supplementary Fig. 2 has been added in the revised version to further substantiate our claim.

3. Page 6, the last sentence of the second paragraph: “Altogether, sun1 sun2 meiotic centromeres displayed a behavior closer to that observed in wild-type somatic cells”. It would be better to add the wild-type somatic cell data to Fig 4, so that it’ll be more straightforward to compare sun1 sun2 with wild-type somatic cells, and then say they displayed similar behavior.

Figure 4 has been modified as suggested by the Reviewer. Average speed has been added for wild-type somatic cells in Figure 4b. For the sake of consistency within the different panels of this figure, we have also added a corresponding view of tracked centromeres in Fig. 4a and the corresponding histogram of turning angles in Fig. 4c. A new “Supplementary Fig. 4” figure has been added to show average speeds in meiocytes and somatic cells in all mutants.

4. Page 6, the third paragraph: “We combined the SUN::GFP markers with a nucleoporin (NUP54) fused to RFP and to the GFP::CENH3 marker.”

It might be unclear for the readers why you used nucleoporin (NUP54). Could you explain the function of nucleoporin (NUP54) and purpose of using this nucleoporin marker in the experiment?

We chose NUP54 to detect nuclear pores because Tamura et al. (10.1105/tpc.110.079947) showed that it can be tagged by a fluorescent protein without losing its association to the nuclear envelope, and also because it is one of the smallest components of the *A. thaliana* nuclear pore, making it easy to express under its own promoter. We clarified this point in the revised manuscript.

5. Page 6, at the end of the third paragraph: “Live acquisitions on male meiocytes expressing the three markers GFP::CENH3, NUP54::RFP, and SUN::GFP revealed that centromere movements (GFP::CENH3 signal) do not correlate with any detectable displacement of the NE markers (Fig. 5B, and SupData_7)”

When I look at Fig 5 and SupData_7, I found it’s difficult to recognize the 5 centromere signals because SUN and CENH3 are both in GFP. When the centromeres move close to NE, it is difficult to distinguish them from each other. The lack of REC8 signal makes it uncertain if the CENH3 signal is indeed on chromosomes or just background. I suggest doing immunolocalization with REC8/SUN/CENH3 and REC8/NUP153/CENH3 antibodies, to have a better view of CENH3 and SUN and better support the conclusion that centromere movements do not correlate with any detectable displacement of the NE markers.

In that part of the manuscript, we asked a very precise question which is to determine if the centromere RPMs we detect and quantify are associated, or not, with a global

rotation of the nucleus, as it has been reported for example in *Drosophila* (doi:10.1038/ncb3249). To answer that question we cannot use fixed material, as suggested by the reviewer, because we need to capture the dynamics of the system. That is why we chose to analyze concomitantly, in live imaging, centromeres and nuclear envelope markers. The reviewer is perfectly right that in these experiments, the centromere identification is not always easy (because CENH3 and SUN are both tagged with the same fluorescent protein). However, even with this limitation, we found that these acquisitions were sufficient enough to answer the question we asked. Indeed, while we can detect centromere displacements (as testified by the centromere tracking provided in Figure 5B, last panel and movie in Supplementary Movie 7), the nuclear envelope remains still: the two nuclear envelope signals (NUP54-RFP and SUN2-GFP) that form crescents at that stage, remain at the same position over the whole acquisition time. We perfectly agree with the reviewer that the objective and design of this experiment were not clearly stated and we have modified the text accordingly.

6. In Fig 6B: there are two other items that need to be improved

1) As shown in Figure 6Bi, in zygotene, it's difficult to recognize all the chromosome ends by just looking at the pictures, which affects the accuracy of telomere number quantification. As it has been proposed that synapsis is typically initiated in chromosome end regions in some plant species, if it's also the case in *Arabidopsis*, I suggest using early loading of ZYP1 or HEI10 as markers for synapsis initiation sites, which would correspond to the chromosome end/telomere regions. You could choose the cells at early zygotene which just start to synapse and have a little bit loading of ZYP1 or HEI10 on the chromosome ends, those sites can be recognized as synapsis initiation sites and telomere regions.

This is a very good point raised by Reviewer 2 and it's absolutely true that synapsis at telomeres helped with the chromosome end identification. This is why, when possible, we have used not only ASY1 and/or REC8 but also ZYP1 or HEI10 (Figures 6, 8, and 9).

2) The HEI10 signal is so weak, could you make it brighter, and put separate HEI10 signal so that the synapsis initiation sites can be recognized more easily?

Sorry for that. We tried to improve the intensity of the HEI10 signal on Figure 6 but we have to keep in mind that HEI10 signal is less strong and less homogeneous than ZYP1 one. That could explain why the reviewer finds the HEI10 signal so weak. Also, we would like to emphasize that the purpose of this figure is to describe the changes in SUN patterns according to meiotic stages. Here, the HEI10 signal is mostly used to stage meiotic cells (HEI10 is absent in leptotene, partial and linear in zygotene, complete in pachytene and forming foci in diplotene). We have included this information in the legend to enhance clarity for the readers. We also show the individual channels in SupData_8A (now Supplementary Fig. 6), and we have added two new Supplementary

Data (Supplementary Movies 8 and 9) that show the 3D reconstruction of the leptotene and zygotene meiocytes shown in Figure 6B. We think it will be helpful to visualize the zygotene synapsed regions better.

Minor points:

1. Page 3, “Average speed measurements confirmed the absence of specific movement at the leptotene stage, as we did not observe a difference between meiotic (55 ± 20 nm/s, $n=130$ tracks) and somatic (46 ± 17 nm/s, $n=34$) nuclei (Fig. 2C, SupData_4). This contrasted with the significant difference observed at the subsequent zygotene and pachytene stages, where the average speed of 106 ± 42 nm/s ($n=96$) and 100 ± 36 nm/s ($n=96$) exhibited a nearly twofold increase.....”

All the speed units here are nm/s, but in Fig 2 and subsequent paragraphs, the speed units are $\mu\text{m/s}$, could you change all the speed units to $\mu\text{m/s}$, to be consistent in the whole article?

Yes, absolutely, sorry for that discrepancy. We have decided to use nm units throughout the figures and text and made the changes accordingly.

2. Page 4, second paragraph: “indicative of frequent backward steps that suggested that somatic centromeres were constrained to remain within a limited domain” Please correct the overuse of “that”

We changed the sentence for “At all stages, the TA distribution for somatic centromeres showed an excess near 180° (Fig. 2D), with approximately 60-65% of deflection angles above 90° , indicative of frequent backward steps and suggesting that somatic centromeres were constrained to remain within a limited domain”

3. Page 7, the second paragraph: “Interestingly, we observed that in the vast majority of the cases, SUN clumps correspond to regions of the NE where chromosome ends are anchored (Fig.6B, SupData_8)”

How do you know these are chromosome ends? It is quite difficult to identify them in Fig 6B1. Do you have probes or antibodies for chromosome ends?

Here, chromosome ends were identified based on the chromosome axis and SC markers (REC8 and HEI10 signals), as we did for the telomere mapping provided in Figures 8 and 9. The only difference is that we have less markers of the chromosomes than on Figures 8 and 9 (because on Figure 6, we also label the NE- SUN detection). In consequence, we have less accuracy in telomere identification. Even so, the resolution is good enough to check if SUN aggregates colocalize with some chromosome extremities. We understand that this colocalization is difficult to estimate on maximum intensity projections (as shown in Figure 6B). That is why we now provide in Supplementary Movies 8 and 9, 3D views animations of both cells shown in Fig6B. We think these movies will show much

more clearly how we can detect chromosome ends located at areas of the NE where SUN accumulates. We thank the reviewer for this remark which helped us to clarify the presentation of these results.

4. Page 7, the last sentence: “When we analyzed the data separating early zygotene cells (EZ, less than 50% synapsis) from late zygotene (LZ, more than 50% synapsis) and early diplotene (ED, less than 50% desynapsis) from LD (more than 50% desynapsis) (SupData_9B)”

Please change LD to late diplotene and add LD in the bracket.

The change has been made.

5. Fig 8,

A. In the picture I only see Z, P and D, please spell out zygotene, pachytene and diplotene in the legend or add them in the picture.

As suggested by the reviewer, zygotene, pachytene and diplotene have been spelled out on Figure 8a.

B. How do you know where the boundary of NE (the transparent membrane in Fig 8B) is? Was the transparent membrane created based on some NE signal (such as SUN)?

For the experiments shown on Figure 8, we do not have the SUN signal available (because, to facilitate telomere determination, we favored chromosome markers to the detriment of the NE markers). To determine the position of the nuclear envelope (or to be more precise, the location of the nuclear periphery) we drew the smallest sphere possible that closely surrounds the entire nuclear content, based on the DAPI signal and/or chromosomal markers. The transparent membrane therefore represents an estimation of the nuclear periphery location. Nevertheless, we know from the experiments where the NE signal is available (SUN signal, Supplementary Fig. 6), that this method provides a very good estimation of the NE location since in most of the nuclei, the SUN signals (NE), tightly surrounds the chromosomes, as visualized on the z-slices). We fully agree with the reviewer that this information is necessary for the reader, and we modified the text accordingly (MM, results, and figure legend).

C. In the text you defined telomere clustering as minimum 6 telomeres grouped. Could you make it clear which telomeres are grouped in one cluster in Fig 8C if this can indeed be discerned? Could you also mark the clustered telomeres in the picture?

The reviewer probably refers here to Figure 8d (and not 8c which shows the position of the telomeres in regards to the nuclear periphery). Concerning Figure 8d, the level of the telomere clustering was given by the color code shown on the left: the more clustered the telomeres are, the bluer they will be colored; the less clustered they are, the redder they will be (the mathematical calculations under that color code is given in the

Materials and Methods section). We have modified Figure 8 legend to explain the clustering better and have added on Figure 8d, a dotted line that surrounds the clustered telomeres. We also provide as Supplementary Movies 10-11-12, animations of the 3D views that decompose the telomere analysis process, and show in 3D where are located the clustered telomeres.

In C and D, if they are same cells, why in the middle picture of panel C, there are 4 telomeres on the top right of the cell, but in the middle picture of panel D, there are only 2 telomeres on the top right of the cell? The left picture of panel D also shows one telomere fewer than the left picture of panel C. Could the authors explain this?

The reviewer is right, they are the same cells, but in panel d we removed the four telomeres which are associated with the nucleolus (NOR-containing short arms of chromosomes 2 and 4, recognized because they are embedded in KNOB heterochromatin). This was explained in Methods section and in the main text, but we now explain it also in the figure legend. In addition, the new Supplementary Movies 10-11-12 should clarify that further.

6. Page 8, the first paragraph, the last sentence: “starting from an average of 11 telomeres per cluster at the EZ to 8 at late diplotene (Fig. 8 G, H)”

1) In Fig 8G, at late diplotene, number of telomeres in bouquet in four cells are zero, only one cell has 8 telomeres in one bouquet, the average should be $8/5=1.6$, not 8. Please correct in the text.

Here there might be a misunderstanding. When we discuss the average number of telomeres in a bouquet, we consider only the cells where we have a bouquet. We exclude from the calculation, the cells without clusterization. In diplotene, 7 cells show a telomere bouquet out of 15. For these 7 cells, the average number of telomeres per bouquet is $7,9 \text{ telomeres/ cell } (n=7)$. We have revised the text to make it clearer and give these details in the revised Supplementary Table 1.

2) Please change EZ to early zygotene.

This has been done.

7. Page 14, in the last paragraph:

“After adding 10 μ L of 60% acetic acid, the slide was incubated on a hot block at 45 °C for 1 min., and the cell suspension was stirred with a hooked needle.”

Please delete the period after min.

This has been done.

8. page 16, In the “ Immunofluorescence studies” “buds were digested for 20 min at 37

°C after fixation in 0.3% (w/v) Pectolyase Y-23 (MP Biomedicals), 0.3% (w/v) Driselase (Sigma) 0.3% (w/v) Cellulase Onozuka R10 (Duchefa).”

Confusing expression. It would be better to say: “after fixation in ...(components of fixation buffer), buds were digested for 20 min at 37 °C in digestion buffer [0.3% (w/v) Pectolyase Y-23 (MP Biomedicals), 0.3% (w/v) Driselase (Sigma) 0.3% (w/v) Cellulase Onozuka R10 (Duchefa)]”.

Modifications of the Materials and Methods were brought to clarify this part.

“The gel pads were mounted in Vectashield (Vectorlabs), primary antibodies were incubated for 2 to 3 nights, washes after primary and secondary antibodies were performed in...” Please list the components of blocking buffer for primary antibody incubation.

The composition of the blocking buffer has been added.

9. Fig 2

B. What’s the meaning of dotted line and dashed line? Please explain in the legend. The same thing should be explained in SupData_3

Thank you for noticing the incomplete description of panel 2B in the caption of Figure 2 and SupData_3. Different styles (dotted or dashed) have been assigned to the coloured curves to distinguish between the three individual tracks. The legend has been completed accordingly.

D. What does the radius of the sector mean? Does it mean the proportion of the cells in this angle? Please explain in the legend.

This point was also raised by Reviewer 1. As stated in our answer above, each histogram shows the normalized distribution (total area equal to 1) of the instant turning angles over all the tracks. We have updated the legend to clarify this point.

10. Fig 3B

What’s the meaning of the gray and purple line? Please explain in the legend.

The magenta curve represents the fit of the confined diffusion model to the experimental data, as described in the results section of the text, while the gray envelope represents the standard error of the mean. We have updated the legend with this information.

11. Fig 5B,

In the figure legend, “For each time frame, we show the overlay between the green signals (GFP::CENH3 and SUN2::GFP) and the magenta signal (REC8::RFP).”

I do not see the REC8 magenta signal in the picture.

Thanks for catching that mistake. The magenta signal corresponds to NUP54-RFP, not REC8-RFP. We have made the correction.

12. Fig 9A and B

There are transparent round spheres outside of the chromosome. How do you know this is the position of nucleolus? Is there any marker/antibody to detect nucleolus?

As explained above and shown in Supplementary Fig. 6, the position of the nucleolus is very easy to identify on z-slices from the 3D acquisitions. We have modified the Materials and Methods to explain that better.

13. SupData_1:

In the video you show, the REC8 signal faded away quickly, and you also mention this in the article. How do you recognize meiotic stages without REC8 signal? Based on DAPI or cell shape?

In the live imaging acquisitions we never used the REC8 signal to stage the anthers because it fades away quickly but also because it was not resolved enough. That is why we decided to establish, in the paragraph “Live imaging of meiosis to study prophase chromosome movements”, the correspondence between cell shapes (as visualized under brightfield) and the meiotic developmental stage (DAPI staining after 3/1 fixation of the anther and chromosome spreading). These experiments showed that the cell shape does not allow to discriminate between zygotene and pachytene stages, but is enough to discriminate early prophase (leptotene) from mid prophase (Zygotene/pachytene) or late prophase (diplotene). In Supplementary Tables 4 and 5 we now give the details of all the acquisitions analyzed in this paper, the staging and the staging approach. In the case of the video shown in SupData_1 (now Supplementary Movie 1), we determined the stage based on meiocyte cell shapes, which is why we know it is either zygotene or pachytene but cannot be more precise.

14. SupData_10A (now Supplementary Fig. 7)

1) It seems the zoomed picture is not exactly the part in the square dashed frame in row 1. Could you mark it clear which part in row1 the zoomed picture comes from?

In fact, the zoomed area was properly indicated, but the view angles were different between the global view and the zoom, which explains why it looked different. We provide a modified SupData_10A (now Supplementary Fig. 7) to avoid this problem.

2) On the right, there are two completely identical representation diagrams, please delete one.

The two diagrams are not exactly the same: only the second shows the KNOB area. We now explain it in the legend.

15. SupData_11 (now Supplementary Table 1)

1) The first column: Please don't use abbreviation of the stages, use full name of the stages. If abbreviations are used, please explain them in the legend.

2) The header of the third column: "Clusters" should correct to "Clusters/Number of cells"

3) The fourth column: What is the meaning of numerator (11, 10, 9, 8, 8) and denominator (16)? Please explain it clearly in the header.

We fully agree with the reviewer that SupData_11 (now Supplementary Table 1) wasn't clear enough and we have made a number of significant changes. We no longer separate early zygotene from late zygotene, nor early diplotene from late diplotene, we now use the full names for the stages, we modified the headers, and provided a legend.

16. SupData_12 (now Supplementary Fig. 9 and Supplementary Movies 13 and 14)

Top picture of B: It is a merge of REC8 and ZYP1, but the ZYP1 signal looks very weak, I can hardly see it. Could you increase the intensity of ZYP1 signal in the first and the third picture of B?

We have increased the ZYP1 signal intensity on SupData_12 (now Supplementary Fig. 9), and provide the corresponding 3D movie stacks (Supplementary Movies 13 and 14).

REVIEWERS' COMMENTS

Reviewer #1 (Remarks to the Author):

The authors responded to all comments included in the review and I found their answers satisfactory. In my opinion, the manuscript has been significantly improved and is now more readable.

I have two comments:

1. I understand the authors' argumentation regarding maintaining the use of the term telomere bouquet in the work. However, in such a case, I would suggest providing the definition of “bouquet” adopted by them in the work, either in the introduction or in the results, so that readers understand what the authors mean when they use this term.
2. Currently, video files have no names, only short descriptions. This makes it much more difficult to find the appropriate supplementary movie according to the numbering in the manuscript. The authors (or Nat Comm editors) need to correct this.

Reviewer #2 (Remarks to the Author):

Thanks for the reply. The authors have answered all the questions perfectly. I don't have further questions.

Reviewer #2 (Remarks on code availability):

I found there are no files in this dataset. I couldn't download the dataset and the code.